# From Noise to Narrative: Tracing the Origins of Hallucinations in Transformers

**Praneet Suresh**[*]
Mila - Quebec AI Institute
praneet.suresh@mila.quebec

**Jack Stanley**[*]
Mila - Quebec AI Institute
jack.stanley@mila.quebec

**Sonia Joseph**
Mila - Quebec AI Institute, Meta AI
sonia.joseph@mila.quebec

**Luca Scimeca**
Mila - Quebec AI Institute
luca.scimeca@mila.quebec

**Danilo Bzdok**
Mila - Quebec AI Institute
bzdokdan@mila.quebec

## Abstract

As generative AI systems become competent and democratized in science, business, and government, deeper insight into their failure modes now poses an acute need. The occasional volatility in their behavior, such as the propensity of transformer models to hallucinate, impedes trust and adoption of emerging AI solutions in high-stakes areas. In the present work, we establish how and when hallucinations arise in pre-trained transformer models through concept representations captured by sparse autoencoders, under scenarios with experimentally controlled uncertainty in the input space. Our systematic experiments reveal that the number of semantic concepts used by the transformer model grows as the input information becomes increasingly unstructured. In the face of growing uncertainty in the input space, the transformer model becomes prone to activate coherent yet input-insensitive semantic features, leading to hallucinated output. At its extreme, for pure-noise inputs, we identify a wide variety of robustly triggered and meaningful concepts in the intermediate activations of pre-trained transformer models, whose functional integrity we confirm through targeted steering. We also show that hallucinations in the output of a transformer model can be reliably predicted from the concept patterns embedded in transformer layer activations. This collection of insights on transformer internal processing mechanics has immediate consequences for aligning AI models with human values, AI safety, opening the attack surface for potential adversarial attacks, and providing a basis for automatic quantification of a model's hallucination risk.

## 1 Introduction

Generative AI systems are becoming always more capable, demonstrating remarkable performance on a wide range of tasks, across diverse data modalities. Yet, transformer models can at times confidently misstate facts, invent details in generated content, and engage in confabulated reasoning. These model behaviors occur even when accurate information for the task is abundantly present in the models' training data or even context window. These plausible-seeming yet incorrect outputs are

---

[*]Equal contribution

39th Conference on Neural Information Processing Systems (NeurIPS 2025).

often referred to as "hallucinations", and range from simple factual mistakes to potentially harmful and deceptive or mis-aligned behaviors [26][1][46][41][65].

While it may be trivial to quantify the error rate of a transformer model using various performance benchmarks, working towards a truly mechanistic understanding of model hallucinations, errors, and inherent biases will be unavoidable as we entrust AI systems with increasingly sensitive and consequential real-world tasks.

For the goal of dissecting the units of meaning on which transformer models internally operate, sparse autoencoders (SAEs) have imposed themselves as a widely adopted tool in the mechanistic interpretability community [12][7]. This class of autoencoders now provides the means for disentangling the opaque visceral workings of large language models (LLMs) into human understandable components. Such SAEs learn a mapping function from intermediate transformer model activations to a sparse overcomplete representational space, which has an interpretable basis by nature, that can be used to faithfully reconstruct the latent activations. Recent work has shown that human interpretable features of consequence can be identified in transformer models at scale, from toy models to commercial-scale LLMs, using SAEs [60].

Much of this interpretability research is rooted in the "linear representation hypothesis". It posits that semantic concepts are represented as single-dimensional linear directions in a representation space derived from transformer layer activations [49][18]. While there is some evidence that certain features may be inherently multi-dimensional in SAE embedding activations [19], there is a growing body of research underscoring the validity and utility of these linearly decomposable single-dimensional features [42][51][21][18].

However, several highly important open questions remain: to what extent do these learned representations in transformer models reflect properties of the input data itself, as opposed to inherent biases or structural priors that emerged during model training? How do these internal representations behave when the transformer model is facing uncertainty due to confusing, ambiguous, or noisy inputs?

Using SAEs as a tool to probe the emergent conceptual landscape of transformer internals, we show that pre-trained transformers impose coherent, steerable conceptual structure even on pure noise or randomly shuffled inputs, with the repertoire of activated concepts expanding as input structure degrades, across both images and text. Crucially, we show that patterns of concept activations elicited by the *input prompt* reliably predict hallucinations in the transformer's generated *output*, providing a practical signal for anticipating unfaithful generations and directly linking high-level inferred concepts to model errors. Taken together, our exploratory and interventional experiments yield three key contributions:

1. First, common pre-trained transformer models exhibit a strong form of input-insensitive inductive bias: these systems tend to impose semantic structure on inputs, tying them into learned conceptual webs, even if the model inputs are ambiguous or lack any coherent meaning.

2. Second, this skewing as information trickles through transformer processing layers exacerbates as input uncertainty increases: the more randomness we experimentally introduce in the input observations, the more persistently the transformer model is drawn towards operating on semantic units that align with familiar internal model representations. We show a demonstrable expansion of semantic concepts that are triggered, localized especially to the middle layers of the transformer model, as the coherence of input information degrades.

3. Third, the constellation of concept activations in a transformer model's intermediate processing representations can be used to reliably predict the tendency of hallucinated or unfaithful output. This insight suggests an automatically measurable link between spurious internal feature recruitment of the input and the fidelity of the output produced by these increasingly popular deep learning systems.

## 2   Preliminaries

In this section, we introduce our experimental framework to probe the emergence of semantic structure in transformer models, from layer to layer, by means of SAE training and concept evaluation. To address our research aims, we employ SAEs trained on transformer activations from pure noise

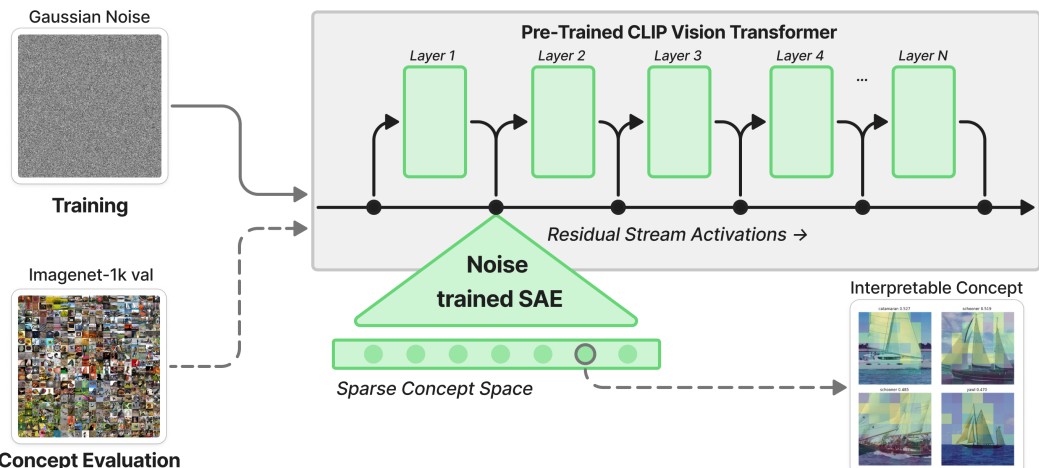

Figure 1: **Workflow to examine hallucination risk in each layer of the transformer.** To assess the extent to which transformer models infer meaningful concepts from semantically void inputs, we study the semanticity of concepts in SAEs trained on residual stream activations of 1.3 million Gaussian noise samples from a pre-trained CLIP vision transformer at each layer by probing the concepts with natural images from the ImageNet-1k validation set. Many concepts are highly interpretable and consistent, despite the SAEs only ever being exposed to pure noise residual stream activations during training. While this noise training setup is specific to our first experiment, subsequent experiments adopt the same concept evaluation approach with alternative transformer models and input modalities. See Preliminaries and Appendix A for further experimental details.

inputs, SAEs trained conventionally from scratch for smaller transformer models, and conventionally pre-trained SAEs for larger transformer models. To ensure that our results are as general and widely relevant as possible, we investigated latent activations from vision and language transformers of varying sizes (detailed in the following subsections and in Appendix A); thus covering several choices of data modality, model size, and model type.

## 2.1 Transformer models

For the vision transformer (ViT) [14], we study the CLIP-ViT-B/32 [53] architecture: the base variant of the vision transformer from OpenCLIP [11], pre-trained on the LAION dataset [57] with 1.4 billion image-text pairs using contrastive supervision. In the case of language transformers [62], we use Pythia-160m-deduped [4], trained on 300 billion tokens from the Pile [22]. We also use Gemma 2B-IT [59] for experiments where coherent text generation and instruction following is required. For all transformer models, the object of investigation is the activations in the residual stream: the hidden state of the deep neural network, updated by attention and multi-layer perceptron (MLP) blocks at each layer. The residual stream is the core conduit for information flow and representation refinement to which layers read from and write to in the transformer architecture [15] [16].

## 2.2 SAE objective

For each SAE formulation, we work with SAEs whose parameters are trained to produce the following approximation for the "residual stream activations", or individual token embeddings $\mathbf{x} \in \mathbb{R}^{d_{\text{model}}}$ for a particular layer:

$$\mathbf{x} \approx \sum_{i}^{d_{\text{SAE}}} f_i(\mathbf{x})\mathbf{d}_i + \mathbf{b},$$

that is, each transformer layer's intermediate activations can be approximated as a (sparse) linear combination of unit direction vectors $\mathbf{d}_i \in \mathbb{R}^{d_{\text{model}}}$ scaled by concept activation coefficients $f_i(\mathbf{x}) \geq 0$, with $\mathbf{b} \in \mathbb{R}^{d_{\text{model}}}$ being the bias term.

Practically speaking, this SAE is a simple neural network with an encoder $F : \mathbb{R}^{d_{\text{model}}} \to \mathbb{R}^{d_{\text{SAE}}}$ projecting each transformer activation into a sparse concept space, typically of much larger dimensionality than the transformer model activation space. For instance, the ViT embedding dimensionality is $d_{\text{model}} = 768$ with the corresponding SAE concept space dimension being $d_{\text{SAE}} = 49,152$. Conversely, the learned decoder $D : \mathbb{R}^{d_{\text{SAE}}} \to \mathbb{R}^{d_{\text{model}}}$ component of the SAE aims to reconstruct the original transformer residual stream activation solely from this pattern of sparse concept activations. This linear decoder $D$ reconstructs the activation, $\hat{\mathbf{x}} = Df(\mathbf{x}) + \mathbf{b}$, whose columns $\mathbf{d}_i$ define the concept directions.

Each SAE in this work is trained end-to-end with the following loss function:

$$\mathcal{L} = ||\mathbf{x} - \hat{\mathbf{x}}||_2^2 + \lambda ||f(\mathbf{x})||_1 \,,$$

where the left (main) term encourages faithful reconstruction (denoted by $\hat{\mathbf{x}}$) of the residual stream activations of the transformer layer at hand, while the right (penalty) term is an $L_1$ regularization constraint encouraging parsimony in the concept space. The composite loss of these two terms motivates the SAE to learn a pruned yet semantically relevant mapping from dense transformer model activations to a sparse semantic concept space. Each SAE is trained independently on a single transformer layer's residual stream activations—e.g., for a 12 layer transformer, we train 12 distinct SAEs on the residual stream activations output from the corresponding layer.

### 2.3 SAE experimental setups

**Noise trained SAEs** For each layer's residual stream probe of the frozen vision transformer from the OpenCLIP ViT-B/32 model, we train a dedicated SAE on activations corresponding to 1.3 million samples of Gaussian noise. Each sample is a 3x224x224 tensor sampled independently from standard Gaussians $\mathcal{N}(0,1)$, clipped to $\pm 3\sigma$ and min-max rescaled to $[0,1]$ before standard ViT normalization.

**SAEs trained conventionally from scratch** To examine overarching principles of the thus decomposed concept spaces of pre-trained transformer models, we also train SAEs on the residual stream activations from each layer of the transformer model under investigation using conventional image and text datasets. We train SAEs on the intermediate activations of OpenCLIP ViT-B/32 from ImageNet-1k [13], and SAEs on the intermediate activations of Pythia-160m-deduped from FineWeb-Edu [50]. For each layer of these transformer models, we train a separate SAE instance; here and in all analogous experiments.

**Pre-trained SAEs** For larger transformer models, we use publicly available SAEs pre-trained on conventional text datasets for comparability with prior interpretability work and to allow analysis on models whose outputs are more realistic, coherent, and representative of practical use cases. In particular, we implement pre-trained Gemma 2B SAEs [5] across layers 1, 7, 11, 13, and 18.

### 2.4 Concept evaluation

To rule out the possibility that the features identified by the noise setting SAE are mere artifacts of the inductive bias of the SAE, we subject each feature in a randomly sampled subset of the SAE features to two complementary, model-agnostic metrics, designed for vision:

1. Semantic purity: To quantify the unequivocality with which a particular feature $f_i$ is present, we collect the top-$k$ ($k = 16$) images that maximize the mean activation of $f_i$ across the sequence of image patches, obtain their CLIP text embeddings (label) and compute the mean cosine similarity (-1.0 to +1.0) of the text embeddings. High semantic purity (close to +1.0) indicates that the images cluster around a single coherent semantic theme. A concept is "pure" if the top-$k$ ($k = 16$ in our case) maximally activating images for that concept have an average cosine similarity of greater than 0.75 across their semantic label embeddings.

2. Steerability: Steerability is the capability of flipping a neutral image's semantic label to that of the concept, by clamping the concept's activation to an artificially high value. To try to "hijack" a particular feature $f_i$ in the transformer model, we add the scaled SAE decoder direction $\alpha \cdot \mathbf{d}_i$ of that feature to the residual stream activations. The scalar $\alpha$ is chosen via a grid search over a pre-defined range of values. We then run the transformer model on feature-naive "neutral" image inputs that originally do not activate $f_i$. If this single vector

addition to a given transformer model layer makes the model's top-1 prediction switch to the feature's own majority label (the concept identified from its images that activate maximally) we consider $f_i$ as "steerable". In our case, conservatively, if $n = 32$ out of 32 images from a randomly sampled batch of neutral images flip to the concept's label after clamping, then this concept is said to be "steerable".

## 2.5 Hallucination

Evaluation of model hallucination is generally recognized to be challenging [26] [41]. As a straight-forward empirical test, we prompt Gemma 2B-IT to summarize a long-form text, and employ the purpose-built and widely used HHEM-2.1 model [3] to provide a continuous hallucination score from 0.0 (no hallucination) to 1.0 (maximum hallucination) for the summary based on its faithfulness to the original text. To directly link hallucinations to our SAE-identified concepts, we fit a partial least squares (PLS) regression layer by layer to predict the hallucination score of the transformer output based on the concept activations of the input prompt. PLS is a linear model that enables interpretable links between concepts and hallucination behavior. To estimate the hallucination score, $h$, we fit:

$$\hat{h} = \mathbf{X}_{\max}\mathbf{W}(\mathbf{P}^T\mathbf{W})^{-1}\mathbf{Q}^T \, ,$$

where $\mathbf{X}_{\max} \in \mathbb{R}^{n_{\text{sample}} \times d_{\text{SAE}}}$ is calculated by taking the maximum concept activations across input tokens for a source, $\mathbf{W} \in \mathbb{R}^{d_{\text{SAE}} \times n_{\text{comp}}}$ is the learned projection matrix into the PLS latent space, $\mathbf{P} \in \mathbb{R}^{d_{\text{SAE}} \times n_{\text{comp}}}$ are the concept loadings, and $\mathbf{Q} \in \mathbb{R}^{1 \times n_{\text{comp}}}$ are the hallucination score loadings. More details can be found in Appendix E.

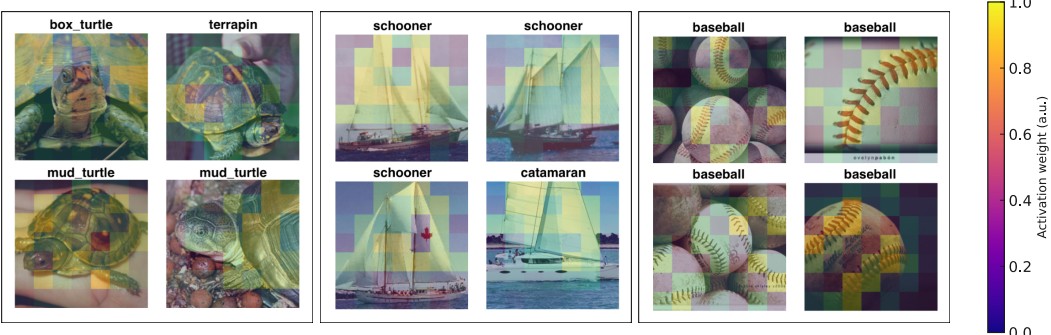

Figure 2: **Semantic concepts are reliably invoked by vision transformer layer 9 activations from pure noise.** Each panel shows the top 4 images among 50,000 candidate images from the ImageNet-1k validation set that most activated a particular semantic concept. These semantic concepts are defined by an SAE trained on pure noise activations from the residual stream at layer 9 of a CLIP vision transformer. Above each image we report the corresponding semantic label for that image. Patch colors indicate the individual patch activation strengths within that image for a given semantic concept (yellow = more activation of the concept). See Appendix B for additional concept examples.

## 3   Transformers represent semantic concepts for pure noise inputs

To investigate the emergence of semantic structure when pretrained transformers are presented with structurally incoherent inputs, we train SAEs on the residual stream activations from pure noise inputs to the CLIP vision transformer. If the pre-trained vision transformer is able to interpret semantic structure from inputs that by construction contain no semantic signal, the ensuing identified semantic concepts could be said to reveal the model's ingrained conceptual biases (Figure 1).

After training one SAE instance per layer on the residual stream activations for 1.3 million Gaussian noise samples, we probe a randomly sampled subset of the noise SAE-identified concepts using 50,000 example images from the ImageNet-1k validation set. To evaluate the interpretability of these concepts, we obtain the top-$k$ ($k = 16$ in this case) maximally activating images (mean concept activation across all patches) for each concept (Figure 2). Additional concept examples from our

noise trained SAEs are presented in Appendix B. We then computed the semantic purity of the concept (cf. Preliminaries) as a proxy for its interpretability, computed by the cosine similarity of the semantic CLIP labels from the top 16 maximally activating evaluation images.

We define a feature as interpretable if its semantic purity is at least 0.75. Using this strict threshold, we observe many high purity features, each corresponding to a coherent high level concept, despite the SAE being trained solely on Gaussian noise activations and never seeing naturalistic image activations during training. We also observe that a subset of these highly interpretable concepts are steerable (cf. Preliminaries): injecting the concept-related activation vector into the residual stream as the transformer model is processing a neutral image can shift the model's semantic label prediction toward the label associated with the concept of interest (Figure 3).

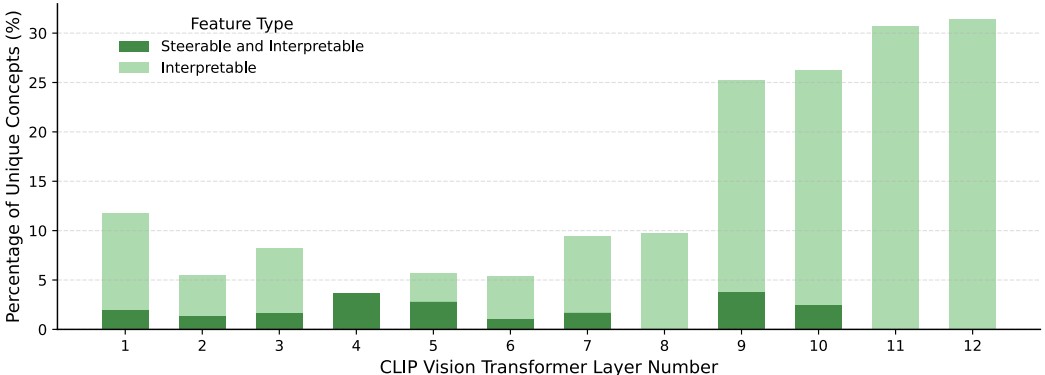

Figure 3: **Transformers fed with noise inputs lead to neuron activations with detectable and controllable semantic structure in many layers.** We measure the interpretability of a concept by the semantic similarity of the labels of the top 16 images that maximally activate that concept. As a more stringent test, we also measure a concept's steerability by the ability of that concept to causally induce its own class label when added to the residual stream of neutral input images. We find that a very large portion of our noise-derived semantic concepts are highly interpretable, and a non-negligible number of these concepts are even steerable, particularly in the early and middle layers. We report the percentage of unique concepts across our noise-trained SAE meeting the aforementioned interpretability and steerability thresholds.

As a further test of concept robustness and layer-wise structuring, we train different noise SAE instances with varying seeds and hyperparameters on the same transformer layer activations. A consistent core of robust concepts [47] emerges across comparable layers. Comparing two sets of noise-setting SAEs reveals a distinctive three-phase pattern in concept overlap, offering insight into how the vision transformer structures information (Appendix C, Figure 16). The first three layers share 35% of their concepts, because the residual stream here still reflects the statistical structure of the raw input, and the space of the viable concepts is highly constrained given the minimal inductive bias at the onset of processing. Conceptual overlap drops sharply to 14% at layers 5-6, marking a stochastic exploration phase within the vision transformer's latent space, exploring a much broader semantic hypothesis space reducing the probability that randomly sampled features from both sets of SAEs will have a large overlap. In deeper layers of the vision transformer model, overlap rebounds to 25% onto a more stable set of high-level features. See Appendix C for further details on these overlapping concepts. In effect, the model settles on the most explanatory subset of concepts for the task, collapsing the expanded mid layer space back into a shared semantic scaffold. Taken together, this three phase pattern indicates that the vision transformer incrementally restructures noise-based activations into coherent task-relevant concepts, and that these emergent representations are sufficiently robust to be discovered by noise-setting SAEs trained with different random initializations.

Grounding our evaluation with the ImageNet-1k validation set gives us a strict, reproducible baseline. Since the concept space is limited by the samples in the validation set, any feature aligned with concepts beyond ImageNet's dictionary is treated as unclassified, making our interpretability and steerability counts understated. A more diverse evaluation set, with images supporting a wider array of ground-truth concepts, would likely show an increase in both of these metrics. Regardless, we have

demonstrated that transformer models ascribe distinct, coherent, and causally actionable concepts to inputs that, by construction, have no semantic content.

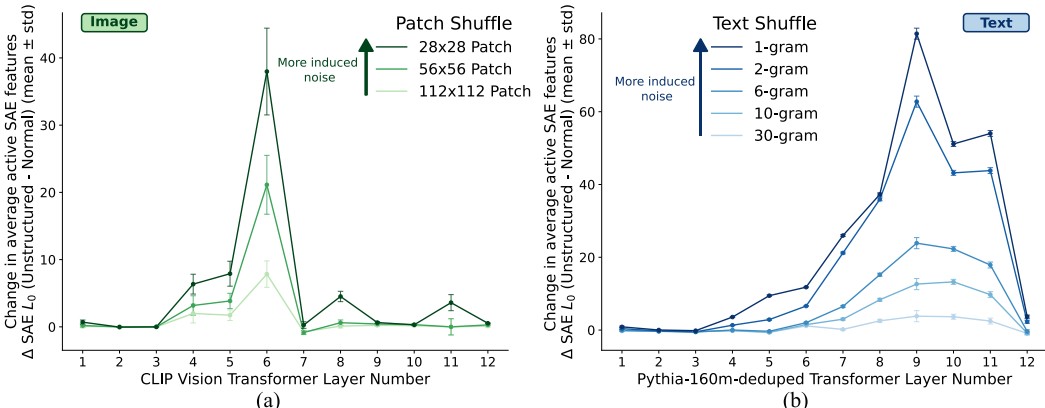

Figure 4: **Increasing uncertainty in vision or text inputs elicits more semantic structure in mid-layers of transformers.** The average number of SAE concepts identified (L0) increases dramatically with increasing input perturbation. We report the average change in L0 from baseline, corresponding to the number of SAE concepts with non-zero activations, across (a) patch-shuffled image activations and (b) n-gram-shuffled text activations for each transformer layer. Error bars show one standard deviation. Smaller patches and lower n-gram count induce greater input uncertainty for images and text, respectively. For both modalities, the L0 difference between natural inputs and perturbed inputs peaks in the middle layers, and increases with increasing levels of deliberate scrambling of transformer input information.

## 4 More unstructured transformer inputs invoke more semantic concepts

To get a better grip on how the transformer models progressively mold the conceptual structure embedded in the input as information flows through the processing layers, we train one conventional SAE instance on each transformer layer independently. These per-layer SAEs are trained on transformer activations from natural images/text, not noise. We provide these normal input activations as a baseline, and then compare the differences in the SAE concept space for varying levels of experimentally induced ambiguity in the input examples. In the case of image inputs to the vision transformer, we randomly shuffle small patches to break the spatial relationship between adjacent parts of the image in natural images (28x28, 56x56, and 112x112 pixel patches shuffled per image), with smaller patches leading to more ambiguous inputs. To this end, for text inputs to the language transformer, we randomly shuffle sets of consecutive words in varying n-gram sizes (single, 2, 6, 10, and 30 word sections), with lower n introducing higher uncertainty in the input space. The transformer layer's ensuing residual stream activations corresponding to these inputs are then fed into the appropriate SAEs for each layer.

A useful macro-level metric for overall SAE concept discovery is the SAE L0: the average number of non-zero concepts across the input sequence activated through transformer processing over the entire dataset. In the case of our image SAEs, remarkably, we notice a steep increase in the average number of concepts activated for randomly shuffled inputs compared to the baseline, peaking at layer 6 with 38 additional concepts activated on average (Figure 4a) for the smallest 28x28 pixel patch size. This pattern holds in the language domain as well. The number of concepts identified per input above the baseline peaks at layer 9 with 81 extra concepts triggered, on average (Figure 4b) for the single word shuffled text. Furthermore, we observe that the degree of input perturbation strongly influences the magnitude of this effect. Input activations with finer-grained shuffling (e.g. single word level or smaller image patches) result in significantly higher L0 count values than input activations shuffled in larger chunks (e.g. 30-word sections or large image patches). See Appendix D for full tables of layer-wise results (Tables 6-7).

These results, converging across different data modalities, are suggestive of the middle layers of pre-trained transformer models playing a more influential role in mediating the model's internal

conceptual instantiation in response to degraded input structure. One might expect transformer models to intuit fewer concepts in shuffled inputs; instead, we find that ambiguity drives an expansion in concept usage, revealing an unanticipated property of how these transformer models organize and impose structure on input processing cascades.

# 5   Input-related internal concept activations predict output hallucinations

We have seen how pre-trained transformer models, when confronted with noisy, uncertain, and ambiguous inputs, appear to traverse the landscape of possible concepts more widely, attempting to impose structure that may not actually be present. Next, we aim to directly link this transformer model-imposed structure in the concept space of the inputs to hallucination propensity with respect to the model-generated outputs. To this end, we first instructed Gemma 2B-IT to produce a generated summary for 1,006 articles from the Vectara hallucination leaderboard [24], employing the purpose-built HHEM-2.1 rating model to give each source/summary pair a real-valued hallucination score, ranging from 0.0 (no hallucination) to 1.0 (most hallucinated). Simultaneously, the transformer's residual stream activations of the original article text were projected into the sparse concept space formed by the pre-trained Gemma 2B SAEs. With these concept vectors in hand, we fit a 4-component PLS on the maximum concept activations across tokens in a sample, attempting to predict the hallucination score of the Gemma 2B-IT-generated text summary.

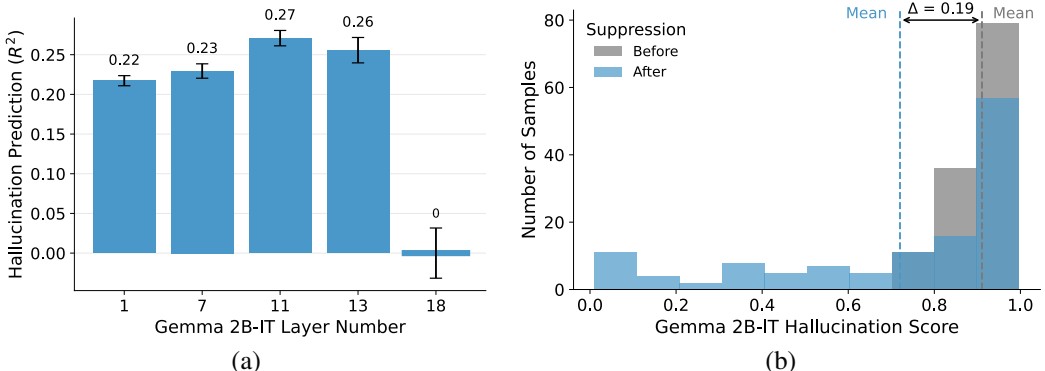

(a)                                                             (b)

Figure 5: **Transformer layer activations can be used to directly predict risk of hallucinated model output.** (a) Hallucination score prediction for the task of faithful summarization of 1,006 source articles. We compare Gemma 2B-IT-generated summaries against the ground truth source articles. We use the sparse SAE concept activations, derived from Gemma 2B-IT residual stream activations, as input to a PLS regression model, predicting the hallucination score for each example. We report 10-fold cross-validated coefficient of determination ($R^2$) on unseen examples, with error bars showing one standard deviation. (b) Suppressing the top 10 SAE concepts in Layer 11's residual stream, identified by the PLS model to be the primary drivers of hallucination, significantly reduces mean hallucination scores across the top quartile most hallucinated examples ($n = 252$). We show a histogram of hallucination scores before (grey) and after (blue) suppression: many examples report significant reductions in hallucination, with a mean score drop of 0.19 in this subset (dashed lines).

Indeed, we find that hallucinations present in the Gemma 2B-IT-generated summaries can be robustly predicted, in unseen text examples, solely from the sparse concept vectors derived from the layer activations of the source prompt using the SAEs. The most accurate predictions are observed for the layer 11 residual stream SAE concepts with a coefficient of determination of $0.271 \pm 0.010$ (out of sample, 10-fold cross validation) for the continuous hallucination scores. For binary classification predicting above versus below the median hallucination score, we reach a mean accuracy of $73.0\% \pm 5.3\%$ across 10 cross validation folds. Transformer layer 18 SAE source concepts do not allow us to predict the continuous summary hallucination scores above chance level (Figure 5a). In Appendix E we also show that our method outperforms existing hallucination detection methods on several additional short form generation and multiple choice hallucination benchmark datasets, despite the fact that we only take into account concepts in the input prompt (Tables 9-12).

Based on this independent mode of investigation, our results reinforce our previous findings (cf. above) that it is especially the intermediate transformer model layers where concept exploration and risk of "conceptual wandering" occur. According to the present analyses, later transformer layers do not appear to participate in this process to a large extent.

We finally explore the practical implications of these findings. Since PLS is a linear model, we can easily reverse-engineer specific key concepts in each transformer layer that have the most influence on the hallucination score predictions. This is because each PLS input dimension corresponds to a distinct SAE-derived concept activation. We select the top 10 key hallucination-associated concepts in layer 11 by variance importance in projection (VIP). With these 10 concepts in hand, we simply suppress them in the residual stream of layer 11 for the input prompt by setting their SAE concept activations to 0, with the SAE decoder yielding hallucination-suppressed residual stream activations. These hallucination-suppressed activations are then swapped in instead of the actual vector of residual stream activations in Gemma 2B-IT's forward pass. Intriguingly, for the top quartile most hallucinated examples in our dataset (252 samples in total), suppressing just these 10 features (out of a total of 16,384) leads to a decrease in the mean hallucination score of 0.19 for this subset (Figure 5b), from 0.91 to 0.72 after suppression. Further details are provided in Appendix E.

This constellation of results shows a clear link between concept space activations of the input prompt and the tendency of the transformer model to hallucinate its generated content. More than this, we present a primitive mechanism to tamp down this tendency to hallucinate.

# 6   Related Work

**Hallucination and perturbed inputs**   The occurrence of hallucinations in generated content is a well-documented issue, persisting across language models regardless of scale or capability [26] [65] [61]. These hallucinations refer to model-generated outputs that are syntactically plausible but factually incorrect or ungrounded to the input prompt. There have been many attempts to benchmark this phenomenon across various tasks [20][24][2][36]. There have been relatively fewer attempts to explain the mechanisms within transformers that underlie these behaviors. [64] attempts to explain (and control) hallucinations based on syntactic errors in particular components of the transformer architecture such as specific attention heads or MLPs. [27] looks at hallucination through the lens of output token dynamics. [32] studies model self-evaluation, and [33] rigorously shows that well-calibrated models will always hallucinate on certain "arbitrary" facts at a baseline rate. These approaches largely employ custom-made datasets and work on the level of single token probabilities and token-wise syntactic dynamics. Our approach links the phenomenon of hallucination directly to interpretable, high-level concepts, and as far as we know is the first study to present a unifying mechanism across both images and text. Our approach also bears some similarities to a large body of research on neural network input perturbation and adversarial examples, where small, often imperceptible changes to the input can lead to erroneous model outputs [58][9][45][63][28][23]. In our work, we introduce highly perturbed inputs as an experimental tool to test the interaction of the actual semantic content of the inputs and the information imposed on the inputs by the transformer.

**Sparse autoencoders**   Early word embedding approaches, such as word2vec [44] and GloVe [52], demonstrated the power of linearly composable embedding spaces for capturing high-level semantic relationships in natural language [43]. More recently, this result has been formalized through the linear representation hypothesis [49], which suggests that many meaningful features of input data can be disentangled and inspected via linear transformations [18][6][17]. Leveraging this insight, sparse autoencoders (SAEs) have proven to be particularly effective tools for breaking down dense high dimensional transformer representations into an overcomplete basis of linear pieces that can be independently studied [12][7][40]. SAEs have surfaced coherent and even manipulable concepts from transformer models of various sizes [60][34]; recent work has extended these techniques to vision transformers with similar success [30][29][25]. There has also been much focus on improving the intelligibility and relevance of SAE-identified features through various alternative training regimes and architectures [21][54][8]. Our experiments leverage SAE-disentangled concepts to localize transformer model behaviors such as hallucinations and conceptual drift, with results that rely solely on the presence of high-level concepts rather than specific SAEs, datasets, or even data modalities. Our approaches are likely to gain strength as interpretability techniques advance.

# 7 Conclusion

We here introduce an objective framework for detecting and profiling structural biases and failure modes in common pre-trained transformer models. Our approach scales organically to transformer models of any size. Crucially, it allows for dataset-independent assessments—eliminating reliance on arbitrary benchmark data subsets and subjective observer-dependent judgements such as highlighting a handful of specific semantic concepts or zooming in on small functional circuits in billion-parameter transformer models. By systematic probing of internal representations as they trickle through the transformer architecture following simple random noise and other controlled input perturbations, we show how to compute a continuous degree of hallucination risk, even on a layer-by-layer basis. We surface presupposed semantic concepts of interest, study the trajectory of their propagation throughout the deep neural network, and examine the nature of how experimentally perturbed inputs become internally represented through fully quantitative, automatically computed measures. In doing so, we reveal important ways in which transformer models are prone to "conceptual wandering", invoking rich semantic structure in inputs with pure noise or degraded semantic information content. Such frameworks may shift interpretability efforts to see deeper into transformer models from qualitative judgements and hand-picked examples to a pipelinable methodology that generalizes across multiple data modalities, expansive model architectures, and experimental setups. Ultimately, our work lays a cornerstone for both real-time monitoring and principled intervention of transformer model behavior, supporting the safe and responsible deployment of emergent AI abilities to the wider society.

**Limitations**   While our experiments span a broad range of model sizes and modalities, it remains to be seen how exactly these findings may manifest in the largest frontier models on the order of 10-100+ billion parameters. Nonetheless, the consistency of observed behaviors across architectures and scales in this work suggests strong potential for generalization. As well, interesting future work would be to explore the interplay of hallucination with increased sequence length and sampling temperature. We would expect increased hallucination and concept activation in these settings. We do not claim that SAEs are perfect oracles for interpretability; rather, our study builds upon the observation that SAEs have been shown repeatedly (in this work and many other works) to extract many useful semantic concepts from transformer activations. Indeed, we have shown here that these concepts can be used to directly predict relevant behavior of model-generated content.

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

# A SAE hyperparameters and training metrics

In Tables 1-3, we report the hyperparmeters used for training SAEs from scratch in this work. In Tables 4-5, we report evaluation metrics for SAEs that we train from scratch in this work.

Table 1: Hyperparameters for OpenCLIP ViT-B/32 SAEs trained on residual stream activations of Gaussian noise inputs. Note that the number of sampled Gaussian noise inputs matches the size of the ImageNet-1k training set, to ensure a fair comparison. Training time was approximately 35 minutes on a single NVIDIA L40S GPU.

| Layer | learning rate | batch size | $L_1$ coef. | $d_{\text{SAE}}$ | $d_{\text{model}}$ | # training images |
|-------|--------------|-----------|------------|-------|--------|-------------------|
| 1 | 3e-3 | 4096 | 2e-6 | 49,152 | 768 | 1.3M |
| 2 | 1e-3 | 4096 | 4e-6 | 49,152 | 768 | 1.3M |
| 3 | 7e-3 | 4096 | 2e-6 | 49,152 | 768 | 1.3M |
| 4 | 2e-3 | 4096 | 2e-6 | 49,152 | 768 | 1.3M |
| 5 | 9e-4 | 4096 | 2e-6 | 49,152 | 768 | 1.3M |
| 6 | 3e-3 | 4096 | 1e-6 | 49,152 | 768 | 1.3M |
| 7 | 7e-3 | 4096 | 1e-6 | 49,152 | 768 | 1.3M |
| 8 | 6e-3 | 4096 | 1e-6 | 49,152 | 768 | 1.3M |
| 9 | 1e-2 | 4096 | 3e-6 | 49,152 | 768 | 1.3M |
| 10 | 3e-4 | 4096 | 1e-5 | 49,152 | 768 | 1.3M |
| 11 | 1e-4 | 4096 | 1e-6 | 49,152 | 768 | 1.3M |
| 12 | 1e-4 | 4096 | 1e-6 | 49,152 | 768 | 1.3M |

Table 2: Hyperparameters for OpenCLIP ViT-B/32 SAEs trained on residual stream activations of natural ImageNet-1k images. Training time was approximately 45 minutes on a single NVIDIA L40S GPU.

| Layer | learning rate | batch size | $L_1$ coef. | $d_{\text{SAE}}$ | $d_{\text{model}}$ | # training images |
|-------|--------------|-----------|------------|-------|--------|-------------------|
| 1 | 6e-04 | 4096 | 1e-1 | 49,152 | 768 | 1.3M |
| 2 | 1e-02 | 4096 | 3e-1 | 49,152 | 768 | 1.3M |
| 3 | 2e-03 | 4096 | 4e-1 | 49,152 | 768 | 1.3M |
| 4 | 1e-04 | 4096 | 2e-8 | 49,152 | 768 | 1.3M |
| 5 | 2e-04 | 4096 | 3e-8 | 49,152 | 768 | 1.3M |
| 6 | 3e-04 | 4096 | 1e-8 | 49,152 | 768 | 1.3M |
| 7 | 2e-03 | 4096 | 1e-8 | 49,152 | 768 | 1.3M |
| 8 | 7e-04 | 4096 | 1e-8 | 49,152 | 768 | 1.3M |
| 9 | 1e-02 | 4096 | 3e-8 | 49,152 | 768 | 1.3M |
| 10 | 1e-02 | 4096 | 9e-8 | 49,152 | 768 | 1.3M |
| 11 | 1e-02 | 4096 | 4e-10 | 49,152 | 768 | 1.3M |
| 12 | 1e-02 | 4096 | 3e-7 | 49,152 | 768 | 1.3M |

Table 3: Hyperparameters for Pythia-160m-deduped SAEs trained on residual stream activations of natural FineWeb-Edu text. Training time was approximately 4 hours on a single NVIDIA L40S GPU.

| Layer | learning rate | batch size | $L_1$ coef. | $d_{\text{SAE}}$ | $d_{\text{model}}$ | # training tokens |
|-------|--------------|-----------|------------|-------|--------|-------------------|
| 1 — 11 | 5e-5 | 2048 | 5 | 24,576 | 768 | 204.8M |
| 12 | 5e-5 | 2048 | 2 | 24,576 | 768 | 204.8M |

Table 4: Evaluation metrics for OpenCLIP ViT-B/32 SAEs trained on residual stream activations of natural ImageNet-1k images.

| Layer | L0 | explained variance | recon. cos sim. | % alive features |
|---|---|---|---|---|
| 1 | 989.19 | 0.99 | 0.99 | 100.00 |
| 2 | 757.83 | 0.99 | 0.99 | 45.39 |
| 3 | 1007.89 | 0.99 | 0.99 | 97.93 |
| 4 | 935.06 | 0.99 | 0.99 | 100.00 |
| 5 | 965.15 | 0.99 | 0.99 | 100.00 |
| 6 | 966.38 | 0.99 | 0.99 | 100.00 |
| 7 | 1006.62 | 0.99 | 0.99 | 99.97 |
| 8 | 984.19 | 0.99 | 0.99 | 100.00 |
| 9 | 965.12 | 0.99 | 1.00 | 92.37 |
| 10 | 854.92 | 0.99 | 1.00 | 85.43 |
| 11 | 1141.99 | 0.99 | 1.00 | 100.00 |
| 12 | 829.09 | 0.99 | 1.00 | 55.71 |

Table 5: Evaluation metrics for Pythia-160m-deduped SAEs trained on residual stream activations of natural FineWeb-Edu text.

| Layer | L0 | explained variance | recon. cos sim. | % alive features |
|---|---|---|---|---|
| 1 | 29.50 | 0.92 | 0.97 | 100.00 |
| 2 | 16.18 | 0.85 | 0.95 | 100.00 |
| 3 | 13.25 | 0.81 | 0.93 | 100.00 |
| 4 | 21.41 | 0.84 | 0.91 | 98.84 |
| 5 | 22.31 | 0.77 | 0.90 | 97.35 |
| 6 | 31.98 | 0.72 | 0.92 | 89.58 |
| 7 | 45.89 | 0.79 | 0.92 | 87.28 |
| 8 | 65.53 | 0.82 | 0.93 | 87.80 |
| 9 | 67.55 | 0.81 | 0.93 | 80.48 |
| 10 | 49.12 | 0.75 | 0.92 | 84.20 |
| 11 | 31.57 | 0.70 | 0.92 | 93.95 |
| 12 | 11.03 | 0.71 | 0.98 | 22.34 |

# B    Additional noise trained SAE concept examples

We present additional examples of concepts identified by the SAEs trained on Gaussian noise activations from the residual stream of the OpenCLIP ViT-B/32, at various layers (Figures 6-15). Evaluation is carried out using 50,000 images from the ImageNet-1k validation set. Early layers largely identify concepts associated with low-level visual features and repeating patterns. Later layers identify very coherent and high-level concepts. These figures extend the results presented in Figure 2.

## Concepts from Layer 3 Noise SAE

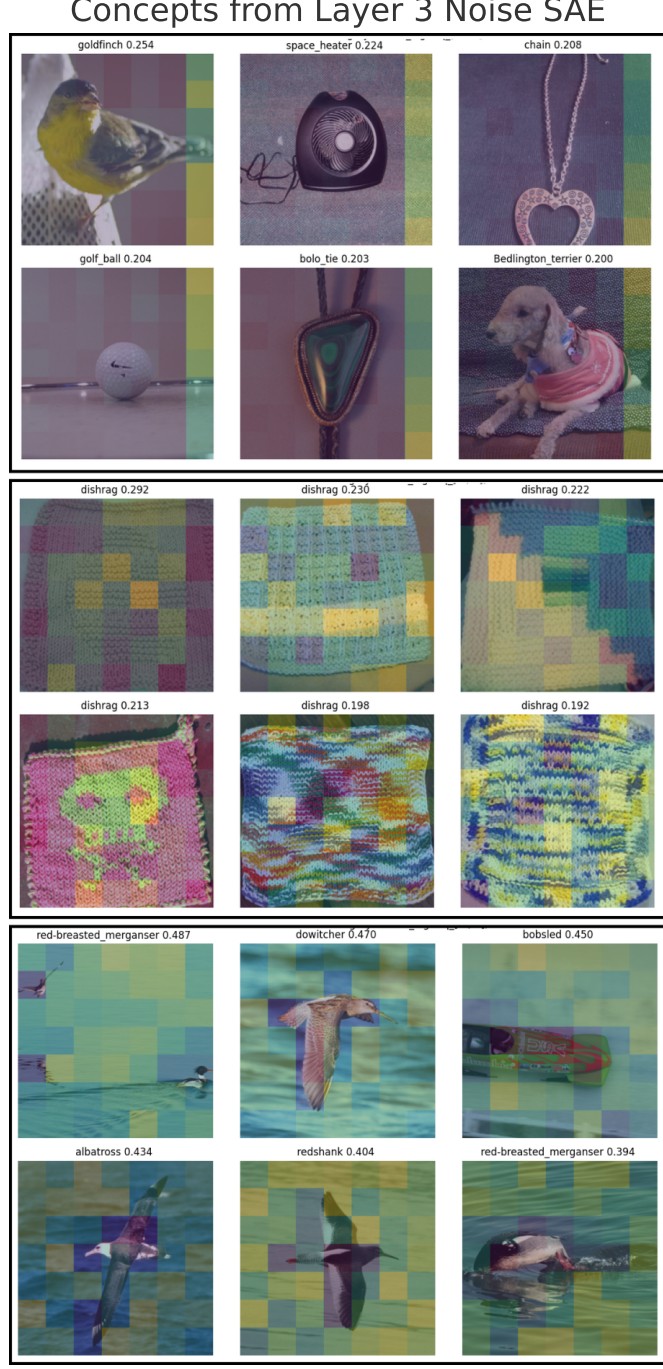

Figure 6: **Maximally activating images for three features from an SAE trained on Layer 3 activations of a ViT with Gaussian noise input**. These features, also representative of early transformer layers (Layers 1 and 2), include: (i) a positional feature that consistently activates for the final patch token across rows, (ii) a texture-sensitive feature that activates for highly repetitive, noise-like patterns resembling dishrags, and (iii) a feature capturing water-like backgrounds with birds in the foreground.

## Concepts from Layer 4 Noise SAE

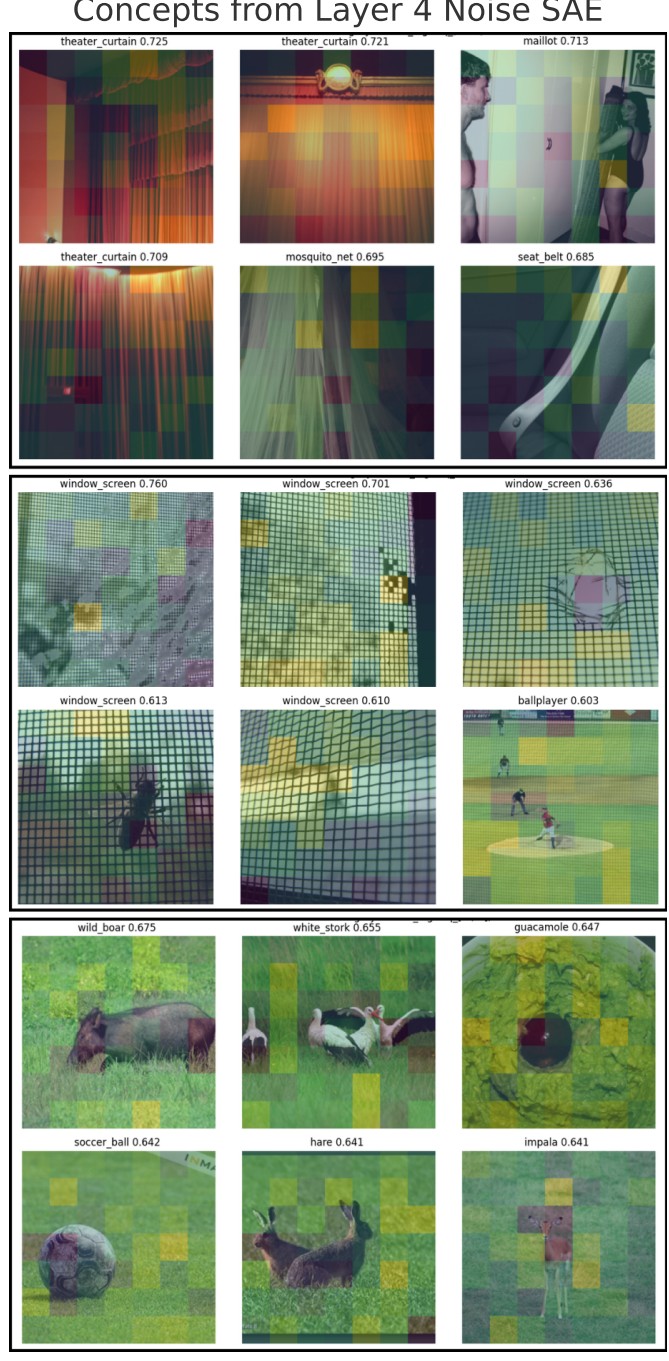

Figure 7: **Maximally activating images for three features from an SAE trained on Layer 4 activations of a ViT with Gaussian noise input**. These features remain grounded in low-level visual patterns but display more variation compared to earlier layers, without forming coherent semantic concepts: i) a feature that activates for vertically folded fabrics such as theater curtains and mosquito nets, likely driven by repetitive linear structures; ii) a feature that responds to mesh-covered textures, such as window shades with overlaid nets, producing a noisy visual appearance; iii) a feature that activates for grassy textures, possibly due to fine-grained spatial repetition.

## Concepts from Layer 5 Noise SAE

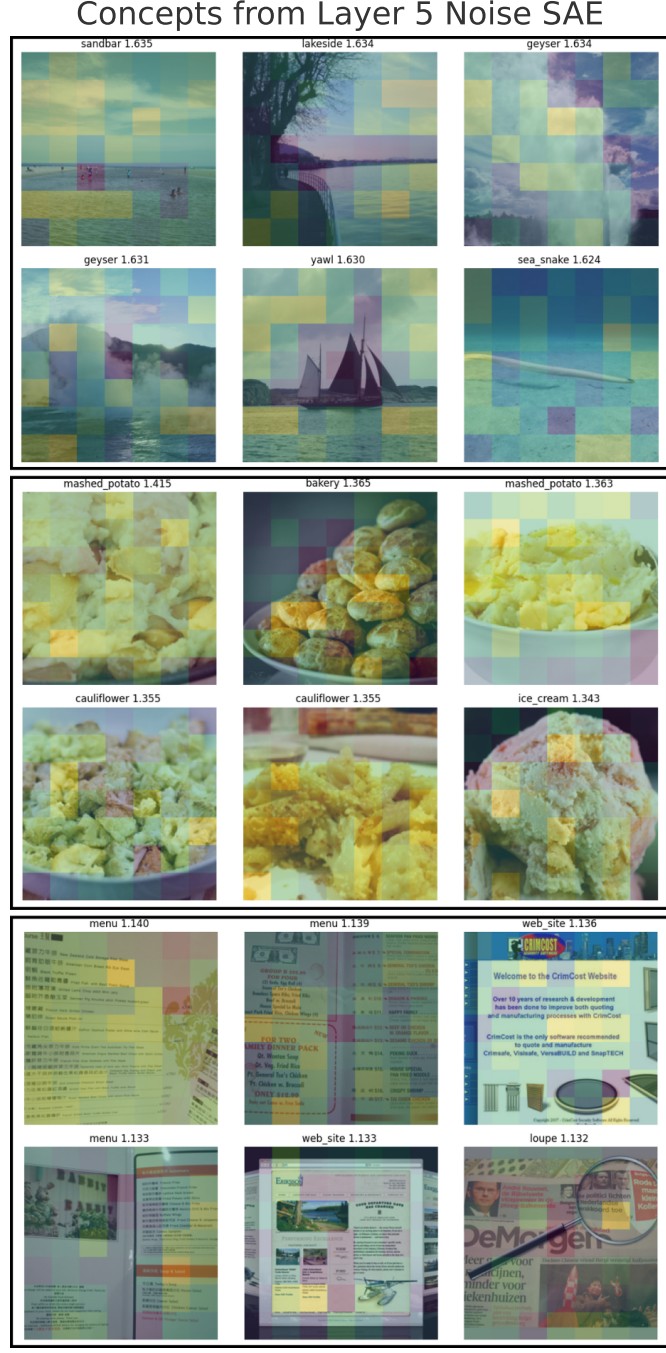

Figure 8: **Maximally activating images for three features from an SAE trained on Layer 5 activations of a ViT with Gaussian noise input**. At this layer, semantic structure begins to emerge, marking a departure from purely textural or position-based features: i) a feature that activates for natural scenes containing clouds and water bodies, indicating sensitivity to environmental layouts; ii) a feature that responds to food items with soft, clumped textures—such as mashed potatoes, ice cream, and cauliflower—highlighting texture-based abstraction; iii) a feature that activates for the presence of printed text, capturing more symbolic and structured content.

## Concepts from Layer 6 Noise SAE

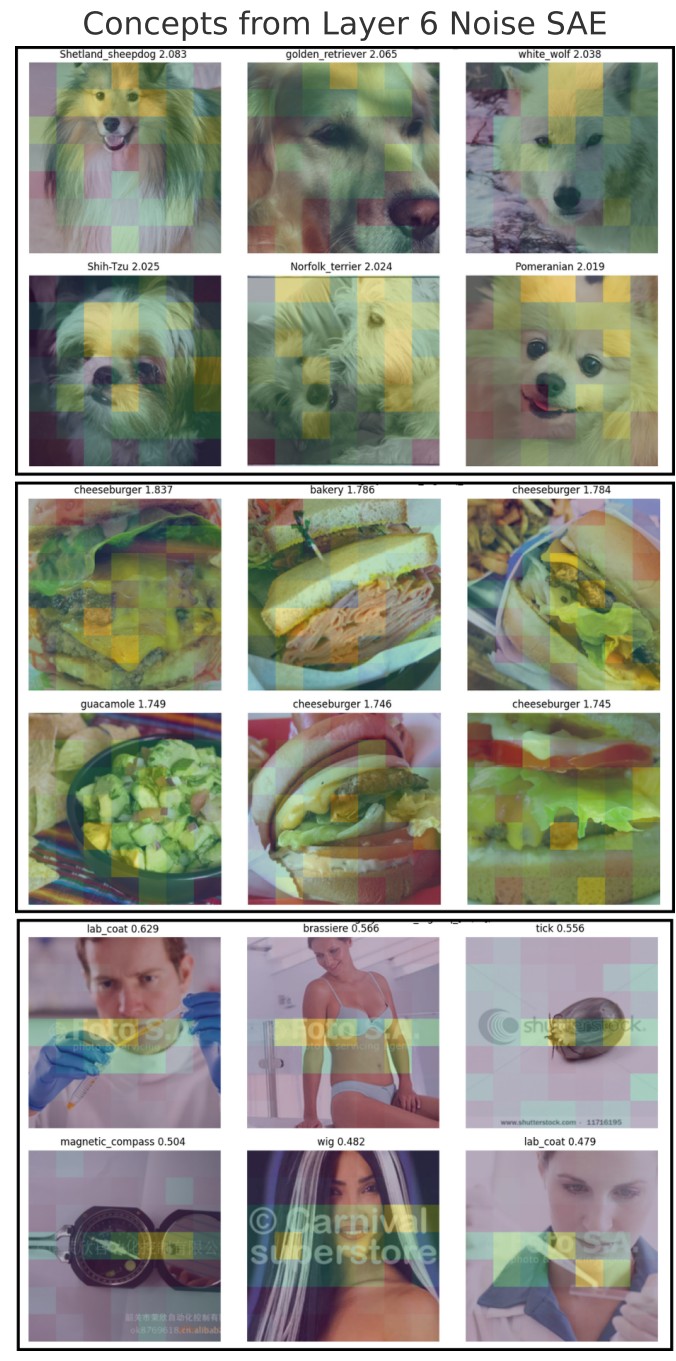

Figure 9: **Maximally activating images for three features from an SAE trained on Layer 6 activations of a ViT with Gaussian noise input**. At this layer, we have features with clear semantics: i) a feature that activates for furry dogs; ii) a feature that activates for food items such as cheeseburgers; iii) a feature that activates for the watermarks on images.

## Concepts from Layer 7 Noise SAE

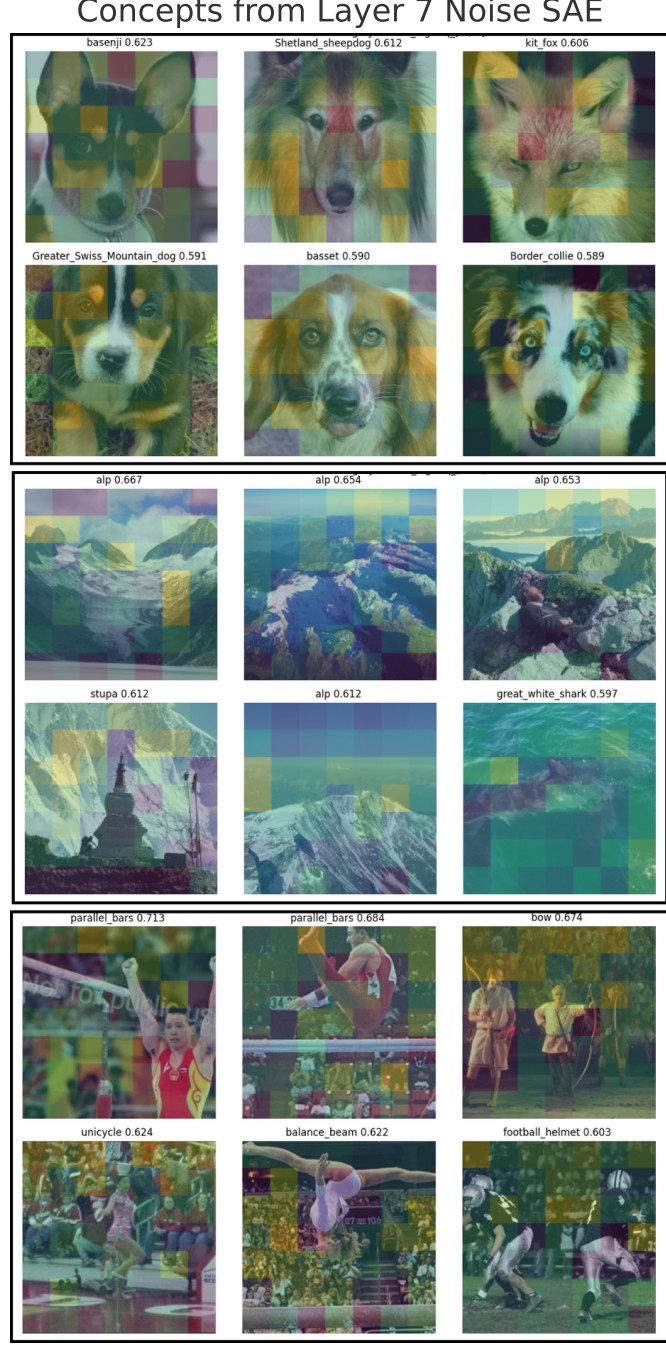

Figure 10: **Maximally activating images for three features from an SAE trained on Layer 7 activations of a ViT with Gaussian noise input**. The features at this depth increasingly correspond to distinct semantic categories and complex scene elements: i) a feature that activates for dogs with upright posture and distinct facial structure, reflecting refined animal categorization; ii) a feature that responds to natural landscapes featuring prominent vertical structures such as mountain peaks, suggesting emerging scene-level abstraction; iii) a feature that activates for human crowds or densely populated scenes, particularly those with small, background-scale human figures.

## Concepts from Layer 8 Noise SAE

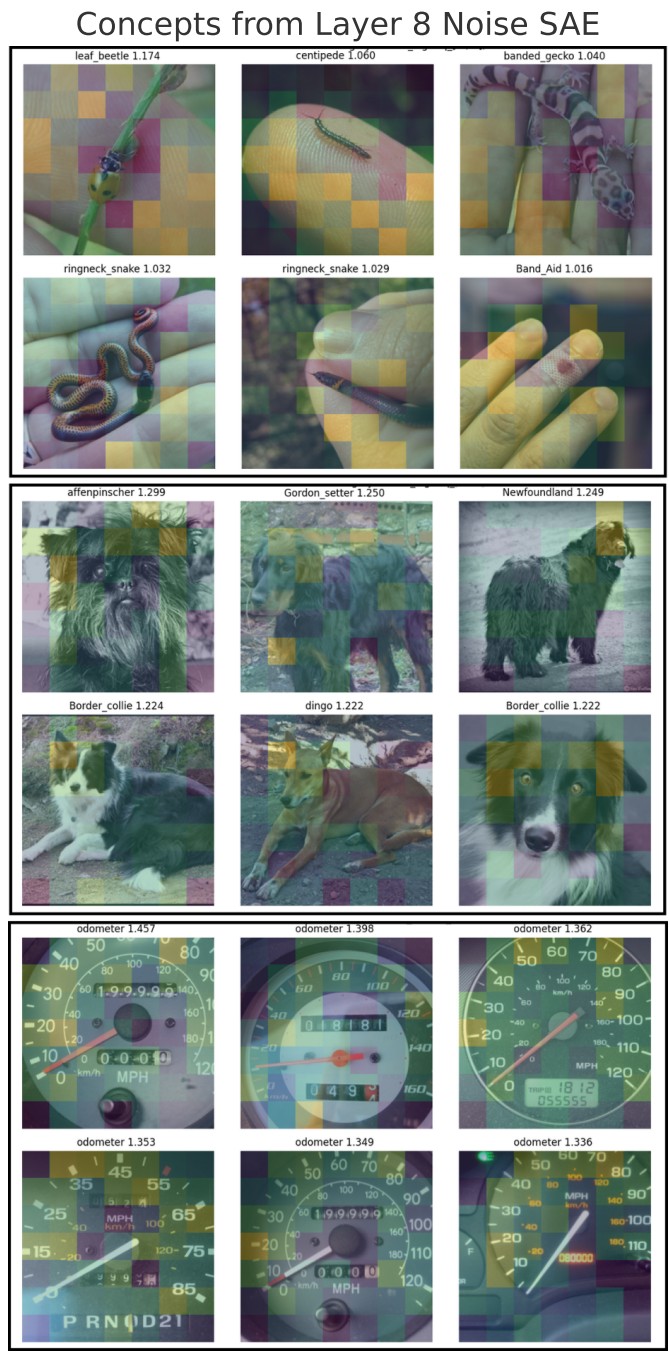

Figure 11: **Maximally activating images for three features from an SAE trained on Layer 8 activations of a ViT with Gaussian noise input**. At this stage, features begin to consolidate around coherent semantic clusters that span both foreground entities and background textures: i) a feature that activates for human fingers, in the context of them being used to hold some organic objects; ii) a feature that activates for dog ears, showing refined selectivity for specific facial features; iii) a feature that strongly activates for odometers.

## Concepts from Layer 9 Noise SAE

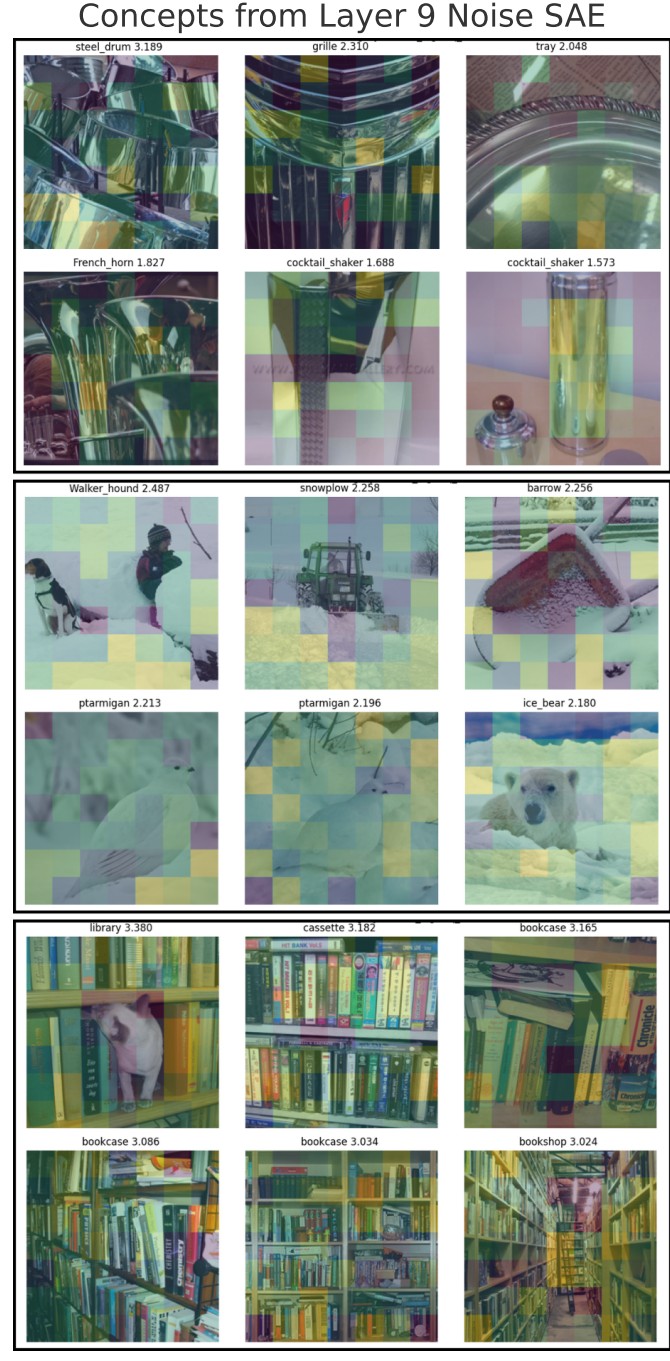

Figure 12: **Maximally activating images for three features from an SAE trained on Layer 9 activations of a ViT with Gaussian noise input**. These features reveal increasingly abstracted and high-level regularities in structured visual scenes: i) a feature that activates for glossy or metallic surfaces with high reflectance, capturing regularities in material properties and lighting conditions; ii) a feature that activates for scenes with snow-covered elements; iii) a feature that activates for bookshelves or library aisles.

Concepts from Layer 10 Noise SAE

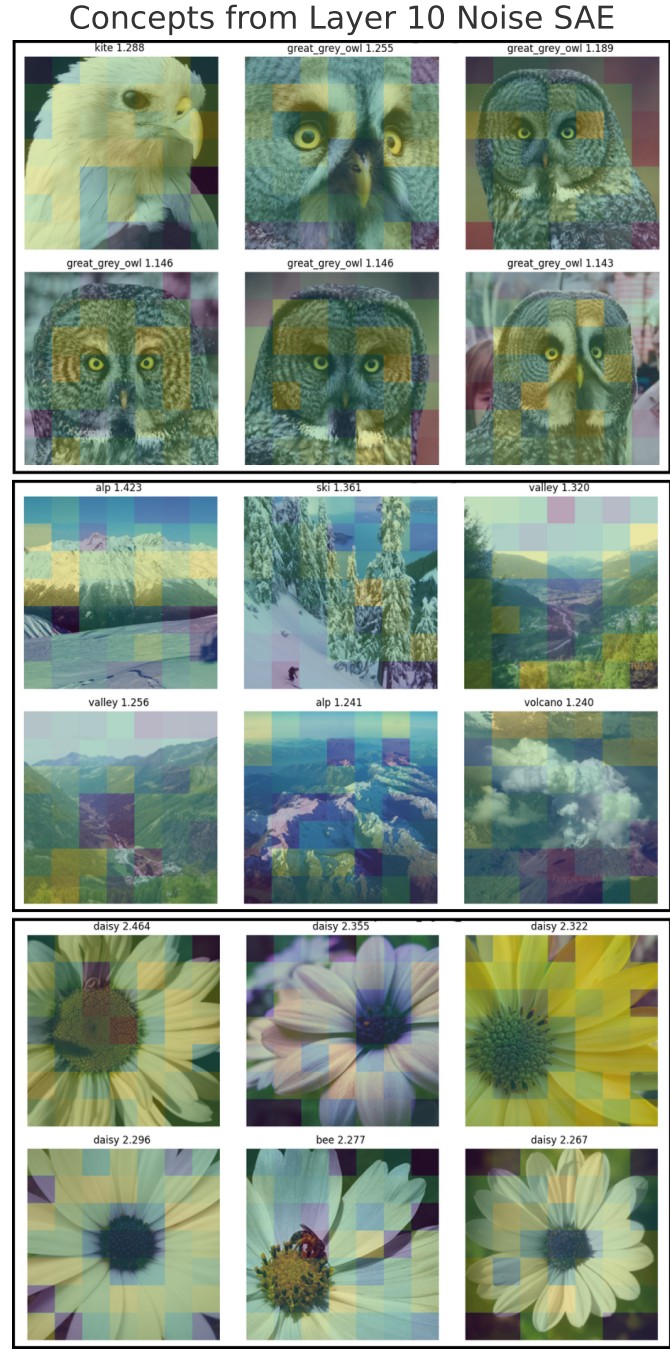

Figure 13: **Maximally activating images for three features from an SAE trained on Layer 10 activations of a ViT with Gaussian noise input**. These features demonstrate increasing semantic clarity and category specificity: i) a feature that activates for close-up views of birds of prey and owls, capturing high-level animal features such as eyes and beaks; ii) a feature that activates for elevated natural landscapes, including snowy peaks, alpine valleys, and volcanoes, emphasizing large-scale geographic structures; iii) a feature that activates for floral structures—especially daisy-like flowers.

Concepts from Layer 11 Noise SAE

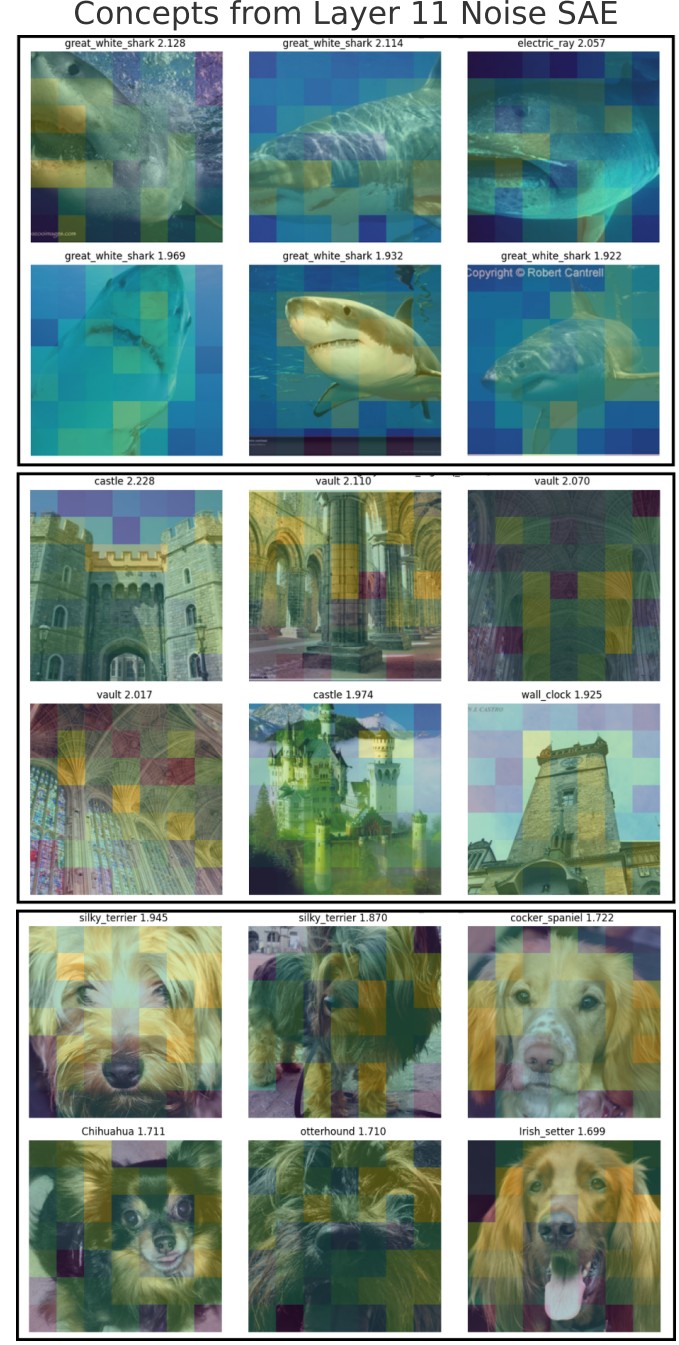

Figure 14: **Maximally activating images for three features from an SAE trained on Layer 11 activations of a ViT with Gaussian noise input**. These features exhibit advanced semantic grouping and fine-grained category emergence: i) a feature that activates for sharks, in different perspectives underwater; ii) a feature that activates for historical architecture and vaulted interiors; iii) a feature that activates for small to medium-sized furry dog breeds.

## Concepts from Layer 12 Noise SAE

Figure 15: **Maximally activating images for three features from an SAE trained on Layer 12 activations of a transformer with Gaussian noise input**. At this deepest layer, we observe strong category-level alignment and rich semantic specificity: i) a feature that activates for jack-o'-lanterns, capturing lighting, shape, and facial carve patterns across varied contexts; ii) a feature that consistently activates for elephants, invariant to pose and background, indicating robust object identity abstraction; iii) a feature that activates for cat faces, spanning multiple breeds and subspecies, reflecting high-resolution encoding of facial features and textures.

## C  Noise trained SAE concept overlap with different initializations

We train two sets of noise SAE instances across all layers with different initialization seeds on the vision transformer residual stream activations derived from Gaussian noise inputs. We then compute the Jaccard index on a randomly sampled feature subset, to estimate the amount of concepts shared between these two sets of SAEs. The Jaccard index is calculated by dividing the number of overlapping concepts by the total number of concepts identified by both SAEs in the subset. For ease of reference, we superimpose the difference in L0 observed for natural image inputs to the L0 observed for maximally shuffled image inputs onto Figure 16 (see Figure 4a for reference). Interestingly, the Jaccard index between the two noise SAEs trained with different random seeds has an inverse pattern to the change in L0 from baseline to unstructured inputs. This provides further evidence that the vision transformer engages in "conceptual wandering" in its middle layers, since the noise SAEs trained from different initializations infer disjoint sets of concepts in the middle layer activations of Gaussian noise. We see this distinct three phase pattern: (i) early layers identify consistent, low-level features, (ii) middle layers widely explore the concept space and impose structure that may not be present in the inputs, and (iii) later layers crystallize the explored concepts into more consistent outputs.

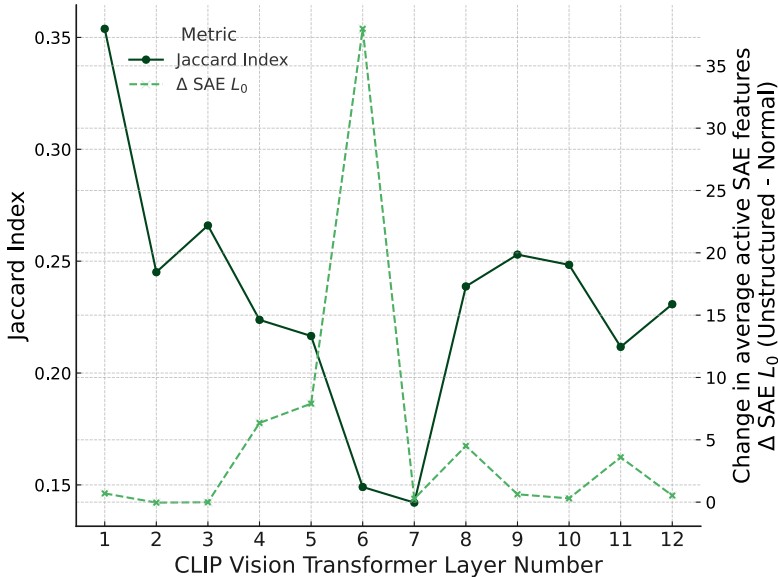

Figure 16: **Conceptual overlap dips in the middle layers of the vision transformer**. We plot the Jaccard index between the concepts identified by SAEs trained on Gaussian noise for two different random initializations (left Y-axis). We have also super-imposed the change in L0 plot for the 28x28 patch shuffled images from Figure 4a for reference (right Y-axis). There is significantly more concept overlap in the early and later layers, which declines in the middle layers. This is an inverse relationship to the increase in L0 observed for patch shuffled images.

## D  Layer-wise L0 for unstructured inputs

We provide mean L0s for varying levels of unstructured inputs, for image and text SAEs. Here we present raw mean L0s and standard deviations in Tables 6-7, as opposed to Figure 4 which shows the change in L0 from the normal image/text baseline.

Table 6: Mean layer-wise L0 for SAE trained on OpenCLIP ViT-B/32 residual stream activations

| | Mean L0 $\pm$ s.d. | | | |
|---|---|---|---|---|
| **Layer** | Normal | 28x28 patch | 56x56 patch | 112x112 patch |
| 1 | $989.19 \pm 0.19$ | $989.89 \pm 0.31$ | $989.38 \pm 0.23$ | $989.27 \pm 0.21$ |
| 2 | $757.83 \pm 0.01$ | $757.79 \pm 0.01$ | $757.83 \pm 0.01$ | $757.83 \pm 0.01$ |
| 3 | $1007.89 \pm 0.03$ | $1007.88 \pm 0.01$ | $1007.88 \pm 0.02$ | $1007.89 \pm 0.03$ |
| 4 | $935.06 \pm 1.17$ | $941.41 \pm 1.47$ | $938.24 \pm 1.44$ | $937.05 \pm 1.40$ |
| 5 | $965.15 \pm 0.28$ | $973.05 \pm 1.85$ | $969.00 \pm 1.13$ | $966.88 \pm 0.78$ |
| 6 | $966.36 \pm 1.02$ | $1004.33 \pm 6.45$ | $987.49 \pm 4.37$ | $974.20 \pm 1.96$ |
| 7 | $1006.62 \pm 0.10$ | $1006.89 \pm 0.48$ | $1005.78 \pm 0.30$ | $1005.89 \pm 0.25$ |
| 8 | $984.19 \pm 0.24$ | $988.70 \pm 0.76$ | $984.80 \pm 0.41$ | $984.30 \pm 0.28$ |
| 9 | $965.12 \pm 0.15$ | $965.75 \pm 0.13$ | $965.54 \pm 0.15$ | $965.34 \pm 0.17$ |
| 10 | $854.91 \pm 0.15$ | $855.21 \pm 0.10$ | $855.23 \pm 0.11$ | $855.08 \pm 0.11$ |
| 11 | $1141.99 \pm 1.20$ | $1145.59 \pm 1.19$ | $1141.99 \pm 1.20$ | $1141.99 \pm 1.20$ |
| 12 | $829.09 \pm 0.21$ | $829.62 \pm 0.05$ | $829.38 \pm 0.10$ | $829.19 \pm 0.16$ |

Table 7: Mean layer-wise L0 for SAE trained on Pythia-160m-deduped residual stream activations

| | Mean L0 $\pm$ s.d. | | | | | |
|---|---|---|---|---|---|---|
| **Layer** | Normal | 1-gram | 2-gram | 6-gram | 10-gram | 30-gram |
| 1 | $29.50 \pm 0.19$ | $30.24 \pm 0.20$ | $29.75 \pm 0.20$ | $29.29 \pm 0.20$ | $29.20 \pm 0.19$ | $29.09 \pm 0.20$ |
| 2 | $16.18 \pm 0.04$ | $16.20 \pm 0.08$ | $15.93 \pm 0.06$ | $15.85 \pm 0.06$ | $15.86 \pm 0.06$ | $15.90 \pm 0.06$ |
| 3 | $13.25 \pm 0.05$ | $13.06 \pm 0.04$ | $12.73 \pm 0.05$ | $12.71 \pm 0.05$ | $12.76 \pm 0.05$ | $12.88 \pm 0.05$ |
| 4 | $21.41 \pm 0.10$ | $24.84 \pm 0.05$ | $22.60 \pm 0.08$ | $21.28 \pm 0.09$ | $21.13 \pm 0.10$ | $21.06 \pm 0.10$ |
| 5 | $22.31 \pm 0.10$ | $31.62 \pm 0.20$ | $25.04 \pm 0.08$ | $21.84 \pm 0.08$ | $21.57 \pm 0.10$ | $21.55 \pm 0.10$ |
| 6 | $31.98 \pm 0.15$ | $43.69 \pm 0.13$ | $38.51 \pm 0.09$ | $33.93 \pm 0.07$ | $33.40 \pm 0.11$ | $33.04 \pm 0.15$ |
| 7 | $45.89 \pm 0.23$ | $72.17 \pm 0.27$ | $67.39 \pm 0.44$ | $52.69 \pm 0.27$ | $49.18 \pm 0.23$ | $46.37 \pm 0.22$ |
| 8 | $65.53 \pm 0.14$ | $102.59 \pm 0.41$ | $101.24 \pm 0.56$ | $80.53 \pm 0.33$ | $73.63 \pm 0.21$ | $67.85 \pm 0.15$ |
| 9 | $67.55 \pm 0.24$ | $149.03 \pm 1.53$ | $130.36 \pm 1.18$ | $91.48 \pm 0.86$ | $80.23 \pm 0.62$ | $71.42 \pm 0.38$ |
| 10 | $49.12 \pm 0.24$ | $100.54 \pm 0.69$ | $92.59 \pm 0.65$ | $71.70 \pm 0.60$ | $62.64 \pm 0.51$ | $53.06 \pm 0.32$ |
| 11 | $31.57 \pm 0.11$ | $85.71 \pm 0.81$ | $75.50 \pm 0.79$ | $49.56 \pm 0.54$ | $41.42 \pm 0.38$ | $34.16 \pm 0.20$ |
| 12 | $11.03 \pm 0.17$ | $14.94 \pm 0.43$ | $13.54 \pm 0.40$ | $10.90 \pm 0.28$ | $10.42 \pm 0.24$ | $10.36 \pm 0.21$ |

Interestingly, using just the L0 from layer 6 of the ViT for each image, we can even predict if the semantic image labels from CLIP will misclassify the image, compared to the ground-truth label from ImageNet. We train a simple linear classifier on L0 counts of SAE latents extracted from 9000 images, and verify the results across 10 independent cross validation folds. We also attempt to use the pattern of concept activations, extracted by a partial least squares (PLS) model, to predict the same misclassification. While still robustly above chance, the results for the PLS model are slightly inferior to the raw L0 results. See Table 8 for the mean AUC scores +/- 1 s.d. on this task. We also test both natural and shuffled images, finding that using the L0 as an input we see similar prediction performance for both types of image. Though the actual rate of hallucination (misclassification in this case) is higher in the shuffled images, it is not easier to predict whether a misclassification will occur for a particular sample.

Table 8: Prediction of semantic mislabeling by CLIP (a proxy for hallucination), using a linear classifier trained only on layer 6 ViT L0 scores for each image as input. Natural and shuffled images tested separately. We report mean AUC +/- 1 s.d. over 10 cross validation folds.

| | **Natural Images** | **56x56 Patch Shuffled Images** |
|---|---|---|
| L0 Misclassification Prediction | $64.8 \pm 1.1$ | $61.3 \pm 1.6$ |
| PLS-Concept Misclassification Prediction | $64.5 \pm 1.5$ | $61.5 \pm 1.3$ |

# E Hallucination task specifications and additional results

We use the popular Vectara hallucination leaderboard [24] evaluation methodology to connect patterns of activated concepts in inputs to hallucination in model-generated output. For this task, we test Gemma-2B-IT by inputting the following prompt:

```
You are a chat bot answering questions using data.
You must stick to the answers provided solely by the
text in the passage provided. You are asked the question
"Provide a concise summary of the following passage, covering
the core pieces of information described." <PASSAGE>\n
```

Where `<PASSAGE>` is replaced with a full length source article or story from the Vectara leaderboard dataset (1,006 examples). This prompt is identical to the prompt used when evaluating models throughout the Vectara leaderboard, and we also set the temperature parameter to 0 during generation to match the task specifications. We use the pre-trained HHEM-2.1 model [3] to give a hallucination score to each model-generated output summary. HHEM-2.1 takes provided a source-summary pair and returns a continuous hallucination score from 0.0 to 1.0, with 0.0 indicating minimum hallucination, and 1.0 indicating maximum hallucination. The motivation for this evaluation of hallucination is straightforward: if HHEM-2.1 detects fabricated content in the summary that is not present the source text, then the summary is said to be hallucinated and is given a hallucination score near 1.0. Note that this task does not measure how factually correct the summary is, rather it only measures the faithfulness of the summary to the source. Therefore this task is well-suited to measuring notions of hallucination and alignment with the task objectives.

With these source-summary pairs in hand, we next fit a PLS regression (see Preliminaries section for details on PLS) on the maximum concept activations across all tokens in the source as given by pre-trained Gemma-2B SAEs [5]. We fit a new PLS model independently for layers 1, 7, 11, 13, and 18 on these concept activations, with the continuous hallucination score for the corresponding model-generated summary as the supervised target. We perform 10-fold cross validation to test the PLS model's generalization on new, unseen example concept activations and hallucination scores. To identify key concept activations in the source that drive hallucinations in the model-generated output (according to the PLS model), we compute variable importance in projection (VIP) scores for each concept. VIP is a metric that pools the contribution of a variable across all PLS components, and is calculated for a single concept $j$ as follows:

$$\text{VIP}_j = \sqrt{\frac{d_{\text{SAE}} \sum_{c=1}^{N_{\text{comp}}} \text{SS}_c w_{j,c}^2}{\sum_{c=1}^{N_{\text{comp}}} \text{SS}_c}},$$

where $d_{\text{SAE}}$ is the number of SAE concepts (in the case of the Gemma-2B SAEs this is 16,384), $N_{\text{comp}}$ is the number of PLS components (4 in this case), $w_{j,c}$ is the weight matrix for projection of the $j$th SAE concept into the latent space of the $c$th PLS component, and $\text{SS}_c$ is the sum of squares for the portion of the hallucination score explained by the $c$th PLS component. Specifically, $\text{SS}_c = q_c^2 (\mathbf{X}_{\text{max}} w_c)^2$ where $q_c$ are the hallucination score loadings for the $c$th PLS component.

To test the practical utility of the most important concepts for hallucination prediction in the PLS, we get the 10 concepts with the highest VIP scores and set their concept activations to 0 during Gemma 2B-IT's summary generation. More specifically, we extract the residual stream activations from Gemma 2B-IT at layer 11, run them through the SAE encoder, and set the resulting SAE concept activations for the 10 VIP-identified concepts to 0. We then run these modified concept activations through the SAE decoder to get reconstructed residual stream activations, and sum these modified reconstructions with the reconstruction error vector for the reconstruction of the unmodified concept activations.

Table 13 lists a few examples of corrected hallucinations post-suppression of the top 10 VIP features in the residual stream of layer 11 of Gemma 2B-IT.

We also provide a comparison of our PLS approach for detecting and classifying hallucinations with existing hallucination detection methods. Tables 9, 10, 11, and 12 show the a comparison of the

results on several different hallucination benchmarks: TruthfulQA (generation) [36], TriviaQA (generation) [31], CoQA (generation) [55], and TruthfulQA (multiple choice). We run each hallucination benchmark on Gemma-2B-IT using "Language Model Evaluation Harness" framework to provide a fair point of comparison. We evaluted on the same 10 cross validation folds. We report the mean AUC +/- 1 s.d. across folds in Tables 9-12. For each benchmark dataset, the hallucination detection method (rows of the Tables) must predict whether new, unseen examples are hallucinated. The "ground truth" hallucination label for each sample is determined according to the metric(s) associated with each dataset (columns of the Tables). For instance, TruthfulQA (generation) evaluates faithfulness with BLEU [48], ROUGE1 [35], ROUGE2, and ROUGEL scores, TriviaQA evaluates correctness with an exact matching string, CoQA (generation) also uses an exact match score and a modified f1 score, and TruthfulQA (multiple choice) evaluates hallucination in settings with either a single correct answer or multiple correct answers.

We find that our PLS and concept-based approach significantly outperforms these established methods on all but the CoQA (generation) benchmark dataset. Further, our method only uses evaluates the input into the LLM, while the other methods require at least some generation of the output text itself.

## F   Impact statement

Our work presents and experimentally tests a hypothesis that transformer model hallucination is connected to an increase in high level concepts inferred by the transformer. We also show that certain types of inputs (in our case noisy or experimentally perturbed inputs) increase the the number of concepts imposed on the inputs by the transformer models. We link the pattern of these high level concepts inferred in the input directly to notions of hallucination in the model-generated output, and show that the hallucination rate of these outputs can be controlled through targeted concept suppression. With this in mind, these hypotheses and techniques could be used to positive (reducing hallucinations, improving output faithfulness) or potentially negative ends (adversarial attacks, misalignment with human values, degraded model performance). Therefore, we expect this work and similar work focused on model alignment and faithfulness to have broad societal impacts in the long run. However, interpretability in transformers and LLMs is still at a relatively nascent stage, and we do not present immediately actionable recipes for negatively controlling or adversarially perturbing transformer models at scale. Thus we do not expect immediate negative societal consequences to stem from this work.

## G   Code availability

Code for the experiments in this paper is deposited at: https://github.com/dblabs-mcgill-mila/noise-to-narrative

Table 9: Comparison of hallucination detection methods on TruthfulQA (generation). Mean AUC +/- 1 s.d. is reported, over 10 cross validation folds, predicting whether an output is hallucinated or not. * = *our method*

|  | BLEU | ROUGE1 | ROUGE2 | ROUGEL |
|---|---|---|---|---|
| Perplexity [56] | 50.0±5.3 | 47.3±5.1 | 49.9±1.5 | 45.7±4.4 |
| Energy Score [38] | 47.2±3.0 | 50.5±3.5 | 49.9±4.5 | 52.4±3.6 |
| Lexical Similarity [37] | 47.0±2.9 | 49.9±2.8 | 53.0±5.7 | 49.1±3.8 |
| Normalized Entropy [39] | 51.1±5.5 | 49.5±4.1 | 49.4±1.6 | 48.6±4.3 |
| EigenScore [10] | 53.9±5.4 | 51.8±4.8 | 47.0±5.9 | 51.0±4.9 |
| PLS-Concepts (Layer1)* | 59.4±6.3 | 56.4±5.0 | 54.4±6.7 | 56.3±5.6 |
| PLS-Concepts (Layer7)* | 59.9±5.0 | 56.6±3.5 | 55.3±7.2 | 54.3±4.6 |
| PLS-Concepts (Layer11)* | **60.0±3.6** | 56.3±4.3 | **58.3±9.1** | 57.3±4.0 |
| PLS-Concepts (Layer13)* | 57.6±5.5 | **58.2±4.7** | 56.6±6.1 | **58.9±5.6** |
| PLS-Concepts (Layer17)* | 54.6±3.7 | 57.0±5.2 | 52.2±4.6 | 54.9±6.2 |

Table 10: Comparison of hallucination detection methods on TriviaQA (generation). Mean AUC +/- 1 s.d. is reported, over 10 cross validation folds, predicting whether an output is hallucinated or not. * = *our method*

|  | Exact Match |
| --- | --- |
| Perplexity | 50.6±2.8 |
| Energy Score | 50.0±0.1 |
| Lexical Similarity | 68.9±2.9 |
| Normalized Entropy | 51.0±1.4 |
| EigenScore | 66.3±2.5 |
| PLS-Concepts (Layer1)* | 66.3±1.3 |
| PLS-Concepts (Layer7)* | 65.6±2.3 |
| PLS-Concepts (Layer11)* | **70.8±2.6** |
| PLS-Concepts (Layer13)* | 70.1±2.4 |
| PLS-Concepts (Layer17)* | 59.1±1.9 |

Table 11: Comparison of hallucination detection methods on CoQA (generation). Mean AUC +/- 1 s.d. is reported, over 10 cross validation folds, predicting whether an output is hallucinated or not. * = *our method*

|  | Exact Match | f1 |
| --- | --- | --- |
| Perplexity | 50.0±0.0 | 50.0±0.0 |
| Energy Score | 50.4±0.4 | 50.1±0.4 |
| Lexical Similarity | **66.4±1.6** | **59.0±2.0** |
| Normalized Entropy | 49.9±0.1 | 50.0±0.1 |
| EigenScore | 62.3±1.9 | 51.0±1.3 |
| PLS-Concepts (Layer1)* | 52.9±1.9 | 52.4±0.9 |
| PLS-Concepts (Layer7)* | 52.6±2.0 | 51.7±1.1 |
| PLS-Concepts (Layer11)* | 53.9±2.6 | 52.1±1.1 |
| PLS-Concepts (Layer13)* | 53.9±1.5 | 51.6±1.3 |
| PLS-Concepts (Layer17)* | 49.4±1.9 | 50.5±1.6 |

Table 12: Comparison of hallucination detection methods on TruthfulQA (multiple choice). Mean AUC +/- 1 s.d. is reported, over 10 cross validation folds, predicting whether an output is hallucinated or not. * = *our method*

| Metric | One Correct Answer | Multiple Correct Answers |
| --- | --- | --- |
| Perplexity | 49.1±4.5 | 49.4±1.9 |
| Energy Score | 50.0±4.2 | 49.8±0.3 |
| PLS-Concepts (Layer1)* | **61.1±2.5** | 62.2±3.7 |
| PLS-Concepts (Layer7)* | 58.9±5.5 | 61.6±3.4 |
| PLS-Concepts (Layer11)* | **61.1±2.6** | 63.2±3.5 |
| PLS-Concepts (Layer13)* | 60.7±3.6 | **63.7±4.1** |
| PLS-Concepts (Layer17)* | 52.4±3.5 | 55.2±4.9 |

Table 13: Gemma 2B-IT recovered hallucination examples post layer 11 suppression of top 10 VIP concepts

| Summaries | | Fabrications | Hallucination Score Change |
|---|---|---|---|
| No Suppression | With Suppression | (no suppression) | |
| The review is expected to be completed by the end of 2023 and will be the first time that a commissioner has undertaken such a comprehensive review of the quality of life and care of older people in Wales. | The passage is about the review of the quality of life and care of older people living in residential care in Wales by the Older People's Commissioner for Wales, Sarah Rochira. | • The article does not mention a completion date
• The article does not claim that this is the first time such a comprehensive review has been conducted | -0.95 |
| Sure, here's a concise summary of the passage: Wenger's contract runs to the summer, and he has been linked with the England manager's job. Southgate has been appointed as England manager and will lead the team against Scotland and Spain in November and December. Wenger is happy that Southgate has a chance to show his quality and believes he has the necessary experience to make the right decisions. Arsenal's Hector Bellerin is set to sign a new contract, but the club is still interested in Barcelona and Manchester City's midfielders, Aaron Ramsey and Thomas Tuchel. | Sure, here's a concise summary of the passage: Wenger's contract runs to the summer, and he has been linked with the England manager's job. Southgate is in contention for the role, and he has been linked to the England national team. Wenger also spoke about the future of Hector Bellerin and Aaron Ramsey. | • Southgate is serving as the interim manager
• The matches are in November, not December
• Aaron Ramsey plays for Arsenal, not Manchester City
• Thomas Tuchel is a manager, not a midfielder, and is not mentioned at all in the original article | -0.82 |
| Jak Trueman's story is a reminder that even in the face of adversity, it is important to stay positive and never give up hope. | **Summary:** Jak Trueman was a 15-year-old schoolboy who was suffering from a rare form of blood cancer called gamma delta T-cell lymphoma. He documented his life and treatment on social media and attracted more than 28,000 Likes on his Facebook page. Despite his illness, Jak remained positive and touched the hearts of many people in West Lothian and beyond. | • Neglects to mention an illness
• Overly vague, misses core facts about the article | -0.74 |

