# OpenReview forum: "From Noise to Narrative: Tracing the Origins of Hallucinations in Transformers"
_NeurIPS.cc/2025/Conference — NeurIPS 2025 poster_

### Official Review · Reviewer_pJz5 · 2025-06-24

**Clarity:** 2
**Significance:** 3
**Originality:** 3
**Rating:** 4
**Confidence:** 3

**Summary:**

This paper investigates how hallucinations arise in transformer models by analyzing residual stream activations using sparse autoencoders (SAEs), where each decoder direction defines a “concept.” Two metrics, semantic purity and steerability, are used to assess these concepts. The authors train SAEs on residual stream from noise inputs to a vision transformer and find that many resulting concepts still align with coherent semantics when tested on real images. They further show that as input uncertainty increases, more concepts are activated—especially in middle layers. Using this framework, the authors predict hallucination rates in Gemma-2B-IT summaries based on concept activations from the input text. The main message of the paper is that increased input uncertainty corresponds to a higher number of activated concepts, which leads to increased hallucination rates.

**Questions:**

See weakness section.

**Ethical Concerns:**

["NO or VERY MINOR ethics concerns only"]

**Final Justification:**

While some concerns remain—particularly the clarity of certain technical definitions and the exploratory nature of parts of the work—I believe the paper presents an interesting and original contribution to understanding hallucinations in transformer models. I am increasing my score to 4 and encourage the authors to further improve the clarity and precision of terminology in the final version.

**Quality:**

2

**Strengths And Weaknesses:**

**Strength:**

1. The use of sparse autoencoders trained on residual stream activations from pure noise is an interesting setup, suggesting that some semantic structure is baked into the model’s internal representations.
2. The study, by varying input (such as image patch shuffling and text n-gram scrambling), is convincing and shows that input ambiguity can lead to higher concept activation of the SAEs.
3. The idea of using internal concept activations to predict hallucination rates in Gemma-2B-IT summaries is a solid step toward connecting the model's internal dynamics to its output behavior.

**Weakness:**

1. While the paper's main goal is to understand the source of hallucination in transformers, much of the work—particularly the analysis focused on vision transformers—feels only loosely connected to this objective. The only part that directly addresses hallucination is the summarization experiment on Gemma-2B-IT. Even there, the results are somewhat limited; it would strengthen the paper if the authors clarified what kinds of hallucinations are being predicted and whether certain types are more reliably captured by the concept activations.
2. The $R^2$ of 0.27 reported in Figure 5 is modest and not quite convincing, and the paper did not compare its method against established hallucination-detection baselines.
3. The paper is difficult to follow at times due to a lack of precise definitions for key terms and notations. For example:
   - how is the function $f_i$ defined?
   - what do you mean by a concept being "activated"? the term was formally never defined.
   - the concepts of semantic purity and steerability are not properly defined.

**Overall:** The paper offers an interesting analysis of how input uncertainty propagates through a transformer’s internal dynamics. However, the step from these observations to a practical hallucination detector and potential mitigation strategies is only loosely argued and supported by limited evaluation. Consequently, I do not believe the work is yet ready for publication at NeurIPS.

---

> ### Author Rebuttal · Authors · 2025-07-31
>
> **[GENERAL]**
> We thank the reviewer for their thoughtful comments and their positive assessment of our experiments. To further support these findings, we have added targeted  experiments with three new text datasets as well as an analogous assessment for the CLIP vision transformers to directly address the reviewer’s points (cf. W1 and W2 below).
>
> **[W1]**
> We recognize that the experiments conducted in this work cover a lot of territory. We first show that transformer models strongly impose semantic structure on inputs, even when they lack true semantic structure (Section 3). This semantic framing is deconvolved from the internal model representations using SAE-identified concepts. We then demonstrate, in images and text, that this semantic imposition increases as inputs become increasingly unstructured and incoherent (Section 4). Finally, we directly link these SAE-identified concepts in the input space of the transformer to hallucinations in its output (Section 5). We desired to test a putative mechanism for the hallucination phenomenon observed in transformers in general, and, after validating this mechanism in images and language, to demonstrate the practical implications for acting on these insights (e.g. model steering of spurious concepts).
>
> As the reviewer points out, there are many kinds of “hallucination” [1]. We tested the “faithfulness” of a summarization task, which is not dependent on the LLM’s background knowledge, as all required information is contained within the prompt. This is therefore a direct test of the effects that the transformer's internal representational structure has on its generated output. For more variety, see the tables presented in W2 for hallucination prediction evaluation on TruthfulQA (Lin et al. 2021), TriviaQA (Joshi et al. 2017), and CoQA (Reddy et al. 2019), as well as comparisons to established baseline hallucination prediction methods. We also perform a new experiment directly linking the L0 and patterns of semantic concepts to the misclassification of image samples (analogous to "hallucination" in the CLIP setting), effectively bridging the vision transformer analyses and the hallucination experiments. Please see the reply to Reviewer e9fj for more details and results (not reproduced here due to space limit).
>
> **[W2]**
> We acknowledge that on its face, an $R^2$ of 0.27 may appear low in some contexts. However, we would like to note that this is a conservative out-of-sample result, averaged over 10 different cross validation folds, predicting a continuous degree of hallucination. This explained variance strengthens our core point that internal layer concept activations can be taken to play a role in output hallucinations. In Section 5, we reported a stronger 73.0% accuracy when classifying examples by whether their hallucination score is above or below the median. Our goal with these experiments was not to build a SOTA “hallucination detector”. Rather, we wanted to show a link between the concept activations and a concrete hallucination metric, and to provide evidence that intervening on these activations can suppress hallucinations to a measurable degree. We show this using a simple linear model. Sophisticated non-linear methods and larger training datasets would likely improve performance. Further, we are predicting output hallucination based on the internal pattern of concept activations derived from the input alone.
>
> We acknowledge the reviewer’s point that additional baselines would improve credibility to our findings and conclusions. We have added the results of several new experiments in comparison with our PLS-Concepts approach. To provide a fair comparison, we run each hallucination benchmark on Gemma-2B-IT using the widely used “Language Model Evaluation Harness” framework (Gao et al. 2024). This framework provides standardized prompt templates and evaluation metrics for many benchmark datasets. Each hallucination prediction method is evaluated on the same 10 cross validation folds, and the mean AUC ± 1 s.d. across folds is reported. For each task, the hallucination detection method (rows of the tables below) must predict whether new, unseen samples are hallucinated, as labeled by the metric associated with each task (columns of the tables below). Note that different tasks evaluate the "hallucination" or "correctness" of an output using different metrics.
>
> *TruthfulQA — Generation*
> ||BLEU|ROUGE1|ROUGE2|ROUGEL|
> |---|---|---|---|---|
> |Perplexity [2]|50.0±5.3|47.3±5.1|49.9±1.5|45.7±4.4|
> |Energy Score [3]|47.2±3.0|50.5±3.5|49.9±4.5|52.4±3.6|
> |Lexical Similarity [4]|47.0±2.9|49.9±2.8|53.0±5.7|49.1±3.8|
> |Normalized Entropy [5]|51.1±5.5|49.5±4.1|49.4±1.6|48.6±4.3|
> |EigenScore [6]|53.9±5.4|51.8±4.8|47.0±5.9|51.0±4.9|
> |PLS-Concepts (Layer1)*|59.4±6.3|56.4±5.0|54.4±6.7|56.3±5.6|
> |PLS-Concepts (Layer7)*|59.9±5.0|56.6±3.5|55.3±7.2|54.3±4.6|
> |PLS-Concepts (Layer11)*|**60.0±3.6**|56.3±4.3|**58.3±9.1**|57.3±4.0|
> |PLS-Concepts (Layer13)*|57.6±5.5|**58.2±4.7**|56.6±6.1|**58.9±5.6**|
> |PLS-Concepts (Layer17)*|54.6±3.7|57.0±5.2|52.2±4.6|54.9±6.2|
>
> *TriviaQA — Generation*
> ||Exact Match|
> |---|---|
> |Perplexity|50.6±2.8|
> |Energy Score|50.0±0.1|
> |Lexical Similarity|68.9±2.9|
> |Normalized Entropy|51.0±1.4|
> |EigenScore|66.3±2.5|
> |PLS-Concepts (Layer1)*|66.3±1.3|
> |PLS-Concepts (Layer7)*|65.6±2.3|
> |PLS-Concepts (Layer11)*|**70.8±2.6**|
> |PLS-Concepts (Layer13)*|70.1±2.4|
> |PLS-Concepts (Layer17)*|59.1±1.9|
>
> *CoQA — Generation*
> ||Exact Match|f1|
> |---|---|---|
> |Perplexity|50.0±0.0|50.0±0.0|
> |Energy Score|50.4±0.4|50.1±0.4|
> |Lexical Similarity|**66.4±1.6**|**59.0±2.0**|
> |Normalized Entropy|49.9±0.1|50.0±0.1|
> |EigenScore|62.3±1.9|51.0±1.3|
> |PLS-Concepts (Layer1)*|52.9±1.9|52.4±0.9|
> |PLS-Concepts (Layer7)*|52.6±2.0|51.7±1.1|
> |PLS-Concepts (Layer11)*|53.9±2.6|52.1±1.1|
> |PLS-Concepts (Layer13)*|53.9±1.5|51.6±1.3|
> |PLS-Concepts (Layer17)*|49.4±1.9|50.5±1.6|
>
> *TruthfulQA — Multiple Choice*
> |Metric|One Correct Answer|Multiple Correct Answers|
> |-------------------------|------------------|------------------------|
> |Perplexity|49.1±4.5|49.4±1.9|
> |Energy Score|50.0±4.2|49.8±0.3|
> |PLS-Concepts (Layer1)*|**61.1±2.5**|62.2±3.7|
> |PLS-Concepts (Layer7)*|58.9±5.5|61.6±3.4|
> |PLS-Concepts (Layer11)*|**61.1±2.6**|63.2±3.5|
> |PLS-Concepts (Layer13)*|60.7±3.6|**63.7±4.1**|
> |PLS-Concepts (Layer17)*|52.4±3.5|55.2±4.9|
>
>
> *\* = our model*
> Based on multiple benchmarks of hallucination classification performance, we note that our method is at least comparable to, if not slightly superior to, many established hallucination prediction tools. We also note that these methods all require some input signal from the LLM-generated content, whereas our method only relies upon concept activations that are internally triggered by the input prompt. As well, Lexical Similarity, Normalized Entropy, and EigenScore require comparisons between multiple LLM generations in order to form their predictions (in the above benchmarks, we set the generation budget to 5). This is why they are not included in the Multiple Choice benchmarks.
>
> We will add these useful results accompanied by a more fulsome description of the experimental details and evaluation protocols in the Appendix.
>
> **[W3]**
> We thank the reviewer for making us aware of the lack of clarity surrounding these terms. We acknowledge that some of these terms could indeed be made more concrete.
>
> [W3.1] In Section 2.2, $f_i(\mathbf{x})$ corresponds to the activation strength of concept i for the input sample $\mathbf{x}$. Thus, the vector $f(\mathbf{x}) \in \mathbb{R}^{d_\mathrm{SAE}}$ corresponds to all the SAE concept activations for the input sample $\mathbf{x}$; we refer to a single element of this vector as $f_i$. In this context, $f$ represents the function of the encoder $F: \mathbb{R}^{d_\mathrm{model}} \to \mathbb{R}^{d_\mathrm{SAE}}$. In response to this comment and the comments from reviewer LyaA, we provide an updated Section 2.2 enhanced for clarity (please see the reply to LyaA for the updated section, not reproduced here due to space limit).
>
> [W3.2] In the mechanistic interpretability literature, “a concept is activated” for a given input if $f_i(\mathbf{x}) \geq 0$ for concept $i$. That is, there is a non-zero entry for concept $i$ in the SAE-encoded vector of the input sample.
>
> [W3.3]
> We define the concept quality metrics semantic purity and steerability more formally in Section 2.4. Semantic purity measures the extent to which the semantic labels associated with image samples that activate for a given concept are similar to one another. A concept is "pure" if the top-$k$ ($k$=16) maximally activating images for that concept have an average cosine similarity of 0.75 across their semantic label embeddings. Steerability is the capability of flipping a neutral image's semantic label to that of the concept, by clamping the concept's activation to an artificially high value. If $n=32$ out of $32$ images from a randomly sampled batch of neutral images flip to the concept's label after clamping, then this concept is said to be "steerable". These conservative metrics are used to validate that the concepts extracted from the noise activation-trained SAE are meaningful. We will update Section 2.4 to reflect the clarified definitions.
>
> **[CLOSING]**
> We believe that these experiments and clarifications significantly strengthen the results and conclusions introduced in our paper. We thank the reviewer for suggesting these improvements.
>
> [1] Huang et al. "A survey on hallucination in large language models" 2023
>
> [2] Ren et al. "Out-of-distribution detection and selective generation for conditional language models." 2022
>
> [3] Liu et al. "Energy-based out-of-distribution detection." 2022
>
> [4] Lin et al. "Towards collaborative neural-symbolic graph semantic parsing via uncertainty." 2020
>
> [5] Malinin and Gales. "Uncertainty estimation in autoregressive structured prediction." 2020
>
> [6] Chen et al. "INSIDE: LLMs' internal states retain the power of hallucination detection." 2024

---

> > ### Comment · Reviewer_pJz5 · 2025-08-04
> >
> > I thank the authors for their detailed responses to my questions and for the additional experiments, which provide additional justifications for the practical implications of the proposed approaches. I am increasing my score to 4 and encourage the authors to define key technical terms more precisely in the main text.

---

### Official Review · Reviewer_e9fj · 2025-06-26

**Clarity:** 4
**Significance:** 4
**Originality:** 4
**Rating:** 5
**Confidence:** 3

**Summary:**

This paper investigates representations of semantic concepts in transformers (in vision and in text) as the uncertainty in the input space is varied through the lens of sparse autoencoders. They find that as inputs become noisier, more semantic concepts are activated. They then attempt to use these findings for the use case of hallucination detection and then steer models to prevent hallucination using the learned concept directions.

**Questions:**

- Is noisy input data even indicative of likelihood of hallucination? Some explanation of how this links to aleatoric/epistemic uncertainty and what kinda of hallucinations you are thinking about would be useful. More experiments from the LLM uncertainty quantification/hallucination literature would also be useful to dissect this. Generalisation to new datasets.
- Do you have an interpretation for why more concepts are activated than fewer? Is there a “randomness”/”noise”/”entropy” concept that emerges for messy input data?
- How do you decompose the likelihood of hallucination as a function of output length, temperature of sampling?

**Ethical Concerns:**

["NO or VERY MINOR ethics concerns only"]

**Final Justification:**

The authors addressed my comments, and I maintain my score.

**Limitations:**

Yes

**Quality:**

3

**Strengths And Weaknesses:**

Strengths:

- Very well written and thorough investigation from first principles from hypothesis to experimental design and results.
- Strong, interesting and (to my knowledge) novel results for mechanistic understanding of concept activations in the presence of noisy data.

Weaknesses:

- The hallucination detection experiments are not particularly convincing as they are lacking detail and not aligned/compared with standard baselines in the LLM uncertainty quantification literature (e.g. short form Q+A generations). Particularly not convinced of the jump from aleatoric noise in input space leading to more concepts translates to hallucinations.
- The macro level results of L0 concept activitations lack the granularity of individual sample results - some qualitative examples on these would be useful.

Overall the paper has lots of exciting insights even without the hallucination section, which needs to be made more concrete.

---

> ### Author Rebuttal · Authors · 2025-07-31
>
> **[GENERAL]**
> We thank the reviewer for their positive assessment of our work and for their constructive comments. We particularly appreciate the reviewer highlighting the novelty and significance of our noise activation-trained SAE experiments and our mechanistic insights into internal transformer concept dynamics. In response to this reviewer comment and the comments from Reviewer pJz5, we have added several new hallucination benchmark datasets and have compared our hallucination detection approach with established baselines. We have also added a new experiment linking the vision SAE-identified concepts to an analogous notion of hallucination in the image setting.
>
> **[W1]**
> We recognize that our hallucination results could benefit from proper contextualization with additional datasets and existing hallucination detection methods. Note that our main goal here was not to develop a state-of-the-art hallucination detection method. Instead, we aimed to reveal a direct link between SAE-identified concepts in the internal representations of transformers and their propensity to produce confabulated outputs. In this sense, the "faithfulness" of summarization is an ideal task as it is agnostic to the internal "knowledge" of the LLM [1]. However, we have conducted additional benchmarks with TruthfulQA [2], TriviaQA [3], and CoQA [4], comparing with established hallucination prediction baselines. These short form Q+A tasks expand the scope of hallucinations studied with our present framework. Please see the reply to Reviewer pJz5 for the full results tables and experimental setup (not reproduced here due to space constraints).
>
> We also recognize that the connection between the noise-input experiments (Sections 2 & 3) and the hallucination experiments (Section 4) could be made stronger. It is somewhat challenging to find exact parallels between vision and language model hallucinations, which is why we focused solely on language hallucinations in the original text. However, to this end, we have conducted additional experiments testing the utility of the L0 metric in predicting an analogous notion of hallucination in the CLIP model: misclassification of semantic image labels. We train a simple linear classifier to predict whether or not an image will be misclassified, based solely on the L0 metric for that image. We source the concept activations and calculate the L0 metric from layer 6 of the vision transformer, as this layer yielded the largest spike in mean L0 differences between normal and shuffled images in Section 4. We train our linear classifier on L0 counts of SAE latents extracted from 9000 images, and verify the results across 10 independent cross validation folds. We find that we can predict whether or not an image will be misclassified solely using its associated L0, robustly above random chance. We also attempt to use the pattern of concept activations, extracted by a partial least squares (PLS) model, to predict the same misclassification. While still robustly above chance, the results for the PLS model are slightly inferior to the raw L0 results (see table below). As well, the natural versus shuffled images seem to yield equal prediction performance; this is likely because although the *rate* of hallucination (misclassification) is higher in the shuffled images, it is not easier to predict whether a misclassification will occur for a given sample. We believe that this new experiment provides a solid bridge between the vision transformer experiments and the direct hallucination prediction tasks with the LLMs.
>
> *Mean AUC (percentage) ± 1 s.d., from ViT layer 6 concepts*
> |  |       Natural Images       | 56x56 Patch Shuffled Images |
> |---|:---:|:---:|
> | L0 Misclassification Prediction | 64.8 ± 1.1 | 61.3 ± 1.6 |
> | PLS-Concept Misclassification Prediction | 64.5 ± 1.5 | 61.5 ± 1.3 |
>
> **[W2]**
> We will add qualitative examples to the Appendix exploring the additional concepts activated for shuffled images over their natural counterparts.
>
> **[Q1]**
> This is an interesting point. Based on the results presented in our paper, we would draw the conclusion that noisy input data is one clear way to increase the likelihood of hallucination. This is evident in the extraneous concept activations that arise as the transformer imposes structure on inputs with little true semantic content (Sections 3 & 4). However, there are likely many other means of inducing hallucinations (such as increased context length, certain topics, etc) [5]. We would consider the experiments presented in this paper to speak only to aleatoric uncertainty. The extraneous concepts activated for pure noise inputs are not a result of "lack of knowledge", but instead a result of the representational systems, shortcuts, and biases instilled in the transformer during its training regimen. This is also why we chose to test hallucinations induced during summarization, as this directly addresses the issue of confabulation despite all the necessary information to complete the task being provided in the prompt. In this way the evaluation of hallucinations is divorced from the inherent "knowledge" of the model and only tests if its internal representational structure of the input has some bearing on the faithfulness of its output.
>
> **[Q2]**
> Our interpretation, which we attempt to put forward in the Conclusion, is that vision and language transformers alike attempt to impose structure on the input data even in the absence of true structure, as a function of inductive biases derived from massive pre-training. We do not notice a specific "noise" concept activated for the shuffled image examples. Instead, we qualitatively see a wide variety of concepts being activated, often with little relation to the true semantic label associated with the natural image.
>
> **[Q3]**
> Thanks, we added a sentence in the conclusion, mentioning these key aspects as targets for future research. We would expect that increases in temperature and context length would be accompanied by increased hallucination, as there is more room for conceptual wandering to occur [6]. However, we note that for the hallucination experiments in the original text we set the temperature to 0 (as stipulated by the Vectara benchmark that we implemented).
>
> **[CLOSING]**
> We are confident that the additional experiments suggested by the reviewer greatly strengthen the robustness and scope of the paper, and provide a more natural connection between the results presented in the main text. We will add these experiments to the Appendix.
>
> [1] Huang et al. "A survey on hallucination in large language models: Principles, taxonomy, challenges, and open questions." 2023
>
> [2] Lin et al. "TruthfulQA: Measuring how models mimic human falsehoods." 2021
>
> [3] Joshi et al. "TriviaQA: A large scale distantly supervised challenge dataset for reading comprehension." 2017
>
> [4] Reddy et al. "CoQA: A conversational question answering challenge." 2019
>
> [5] Kang et al. "Unfamiliar finetuning examples control how language models hallucinate." 2024
>
> [6]Liu et al. "Lost in the middle: How language models use long contexts." 2023

---

> > ### Comment · Reviewer_e9fj · 2025-08-03
> >
> > Thank you for your response, they have addressed my main concerns, and I maintain my score.

---

### Official Review · Reviewer_5gvB · 2025-07-02

**Clarity:** 3
**Significance:** 3
**Originality:** 3
**Rating:** 5
**Confidence:** 3

**Summary:**

This paper describes the use of sparse auto-encoders (SAEs) to probe conceptual representations at different layers of Transformer models, towards a mechanistic understanding of ‘hallucinations’ in these models. Three experimental setups are explored: SAEs trained on the activations of Transformers exposed to pure noise; SAEs trained on activations from Transformers exposed to real image and text data; and pre-trained SAEs.

A key finding of the paper is that Transformer models are prone to imposing semantic structure on meaningless inputs (perhaps akin to the phenomenon of ‘pareidolia,’ or the human tendency to perceive meaningful structures in random stimuli, e.g. seeing faces in clouds). Another finding is that perturbed inputs activate a greater number of concepts than unperturbed inputs in the middle layers of Transformer models, an effect that scales up with the degree of perturbation. Additionally, the findings suggest that hallucinations can be steered based on selective inputs to a model.

**Questions:**

None at this time.

**Ethical Concerns:**

["NO or VERY MINOR ethics concerns only"]

**Final Justification:**

Having considered the other reviews & rebuttals, alongside my original review, I maintain my original score.

**Limitations:**

Yes

**Quality:**

3

**Strengths And Weaknesses:**

Strengths:
The paper is well-organized and clear, with ample definitions and concise descriptions of relevant prior work. The paper provides an interesting and compelling approach to tracing hallucinations in Transformer models, with valuable findings that pave the way for future work in AI safety and interpretability, while complementing existing work. Appendix B offers a particularly valuable case study of how different kinds of features are activated in each layer of OpenCLIP ViT-B/32 exposed to pure noise, with increasing conceptual specificity as layers progress.

Weaknesses:
Not aware of serious weaknesses in the paper. Just an observation: References [35] and [36] are the same paper (Maynez et al 2020).

---

> ### Author Rebuttal · Authors · 2025-07-31
>
> **[GENERAL]**
> We thank the reviewer for their thorough reading of the paper and their enthusiastic appraisal of our work.
>
> **[W1]**
> We will merge references [35] and [36] as per the reviewer’s helpful comment.

---

### Official Review · Reviewer_LyaA · 2025-07-06

**Clarity:** 3
**Significance:** 3
**Originality:** 3
**Rating:** 4
**Confidence:** 4

**Summary:**

At a high-level, the paper shows transformer models create signals (concepts) out of noise. The middle layers contribute more to this aspect, and this contributes to the footprint of hallucination. More specifically, the paper trains a SAE to map residual activations to concepts. The paper presents performs multiple experiments to demonstrate the validity of the approach using both text and image models. (1) Semantic concepts are invoked by pure noise inputs [Section 3]. (2) Distorted inputs also invoke concepts [Section 4] (3) Internal activations are also predictive of hallucinations [Section 5]. All the experiments leverage SAE-induced representations.

**Questions:**

1.	Define residual activation.

2.	What is the significance of 1.3M?

3.	Are d_i in section 2.2 learned or initialized?

4.	In section 2.2, x in LHS is of dimension d_model, but the RHS is of dimensions d_SAE since the unit vectors have that dimension. How does this all fit?

**Ethical Concerns:**

["NO or VERY MINOR ethics concerns only"]

**Final Justification:**

It was my mistake not to see the appendix. This got resolved.

In addition, the authors have provided clear descriptions of the algorithm.

Hence I am increasing my score from 2 to 4.

**Limitations:**

Yes

**Quality:**

3

**Strengths And Weaknesses:**

S1. The problems addressed in the paper (interpreting activation of transformer models) are important and the conclusions (that SAE-induced interpretable representations can be used to trace hallucinations) are plausible.

S2: The paper presents multiple experiments to validate the proposed framework.

W1. The details of modeling are missing in the main paper.  Unfortunately, I am not able to find the appendix sections in the downloaded version. Hence it is impossible to provide an informed review.

W2. The experimental details (such as sizes of datasets and train/test splits) are missing.

---

> ### Author Rebuttal · Authors · 2025-07-31
>
> **[GENERAL]**
> We thank the reviewer for confirming the validity of our experiments and conclusions. We appreciate that the reviewer finds the problems addressed in our paper important, and the conclusions we draw from our experiments to be sound. We also appreciate that the reviewer considers our experiments a proper validation for the mechanistic framework of hallucinations put forward in the paper.
>
> **[W1]**
> We note that the Appendix is included as a PDF named “appendix.pdf” in the “Supplementary Material” zip file attached to the OpenReview submission. Within the Appendix, we have provided full details on model specifications, hyperparameters, and training procedures for each transformer model and SAE used in the paper (Appendix A). In addition to the Preliminaries section in the main text, which describes our modeling setup and motivations for each experiment, we have provided additional details for our hallucination experiments in Appendix E.
>
> **[W2]**
> Directly addressing the reviewer's point, Appendix A provides the number of input image samples used to train the layer-wise vision SAEs (1.3M images from the ImageNet-1k training set), and the number of tokens used to train the layer-wise language SAEs (204.8M tokens from the FineWeb-Edu corpus). In Section 3 we state the number of validation image samples sourced from the ImageNet-1k validation set: 50,000. For better readability, we will add additional columns to the SAE hyperparameter tables in Appendix A corresponding to the size of the validation sets (50,000 image samples in the case of the vision SAEs, and around 5.12M tokens in the case of the language SAEs).
>
> **[Q1]**
> In Section 2.1, we define the “residual stream activations” as follows:
> >For all transformer models, the object of investigation is the activations in the residual stream: the hidden state of the deep neural network, updated by attention and multi-layer perceptron (MLP) blocks at each layer. The residual stream is the core conduit for information flow and representation refinement to which layers read from and write to, in the transformer architecture.
>
> However, we understand how this may be an unclear definition to readers less familiar with the recent mechanistic interpretability literature. More directly, the residual stream activations refer to the token-level embedding vectors extracted from the transformer model following each layer block [1][2][3]. We will add this additional clarification to Section 2.1.
>
> **[Q2]**
> We believe that the reviewer is referencing the number of random Gaussian noise images sampled to train the noise SAE in Section 3. We chose 1.3M as a training set size for this experiment to mirror the size of the ImageNet-1k training set, which is used to train SAEs on natural image inputs for the experiments in Section 4. We will add an extra clarification to the appendix justifying this training set size in the noise setting.
>
> **[Q3]**
> The $\mathbf{d}\_i$ are learned. The $\mathbf{d}\_i$ defined in Section 2.2 refer to the columns of the decoder matrix: $D: \mathbb{R}^{d\_\mathrm{SAE}} \to \mathbb{R}^{d\_\mathrm{model}}$. All parameters of this decoder are learned in conjunction with the encoder, using the same loss signal defined in Section 2.2.
>
> **[Q4]**
> We thank the reviewer for pointing out this oversight on our part. The dimensionality of both $\mathbf{d}\_i$ and $\mathbf{b}$ should be $\mathbb{R}^{d\_\mathrm{model}}$ not $\mathbb{R}^{d\_\mathrm{SAE}}$. With these fixes, the equation for the reconstruction of the activation sample $\mathbf{x}$ is correct, as $f\_i(\mathbf{x})$ is a scalar and we sum over all $i$ SAE dimensions. We will update the paper to reflect this change, as well as clarify the role of the decoder and its training to address [Q3]. Here is the updated Section 2.2:
>
> > For each SAE formulation, we work with SAEs whose parameters are trained to produce the following approximation for a given transformer model activation vector $\mathbf{x}\in \mathbb{R}^{d_\mathrm{model}}$ at a particular layer:
> >
> > $$
> > \mathbf{x} \approx \sum\_i^{d_\mathrm{SAE}}f_i(\mathbf{x})\mathbf{d}\_i+\mathbf{b}\,
> > $$
> >
> > that is, each transformer layer’s intermediate activation can be approximated as a (sparse) linear combination of unit direction vectors $\mathbf{d}\_i\in \mathbb{R}^{d\_\mathrm{model}}$ scaled by concept activation coefficients $f_i(\mathbf{x})\geq0$, with $\mathbf{b}\in \mathbb{R}^{d\_\mathrm{model}}$ being the bias term. Practically speaking, this SAE is a simple neural network with a learned encoder $F: \mathbb{R}^{d\_\mathrm{model}} \to \mathbb{R}^{d\_\mathrm{SAE}}$ projecting each transformer activation into a sparse concept space, typically of much larger dimensionality than the transformer model activation space. For instance, the ViT embedding dimensionality is $d\_\mathrm{model} = 768$ with the corresponding SAE concept space dimension being $d\_\mathrm{SAE} = 49,152$. Conversely, the learned decoder $D: \mathbb{R}^{d\_\mathrm{SAE}} \to \mathbb{R}^{d\_\mathrm{model}}$ component of the SAE aims to reconstruct the original transformer residual stream activation solely from this pattern of sparse concept activations.
>
> **[CLOSING]**
> Experimental details provided in the Appendix, as well as the clarifications of terminology and math notation detailed above should address the reviewer’s outstanding concerns. Thanks to the reviewer, the additional clarity and modeling details will greatly improve the readability of the paper.
>
> [1] Elhage et al. "A mathematical framework for transformer circuits." 2021
>
> [2]Elhage et al. "Privileged bases in the transformer residual stream." 2023
>
> [3] Cunningham et al. "Sparse autoencoders find highly interpretable features in language models." 2023

---

> > ### Comment · Reviewer_LyaA · 2025-08-03
> > **Apologies and Thanks!**
> >
> > My apologies for missing the supplementary section.  I had downloaded the zip file with all the papers to be reviewed. In the past, this zip file had supplementary material but not this time. In addition, a few other papers had supplementary content in the main pdf itself. My apologies again.
> >
> > I thanks the authors for the responding to my questions.
> >
> > The appendix does provide context for the 1.3M number mentioned in the paper (and clarified by the authors in their response). Having said that the appendixes contain details of the PLS model and not the details of the main model. The main section of the paper is not self-contained (and reads somewhat like a blog about a technical paper) and the terminology and details are not specified. This is highlighted by reviewer pJz5 too while the other two reviewers have mentioned that the paper is well-written. I think this is the main drawback of the paper.
> >
> > I am open to revising the scores if the authors are able to provide a crisp and self-contained description of the model.

---

> ### Author Response · Authors · 2025-08-05
> **Self-contained description of modeling pipeline**
>
> We are happy to provide concrete details for our modeling architectures and analysis setup, which we will summarize in our revised NeurIPS paper. We recognize that we could have made more explicit the distinction between the model under study (the transformer) and the experimental tool we use to explore it (the sparse autoencoder).
>
> The “main models” of study in this paper are OpenCLIP ViT-B/32 (vision transformer) and Pythia-deduped-160M / Gemma-2B-IT (language transformer). Note that these transformer models were not trained or finetuned as part of our work, but were instead sourced as pre-trained models from Hugging Face:
>
> | Model | Input Data Modality | Layer Count | Embedding Dimensionality (per token) | Max Context Length | Pre-training Dataset | Pre-training Dataset Size | Pre-training Objective Function |
> | ----------------------- | ------------------- | ----------- | ------------------------------------ | -------------------------------------- | ------------------------------------------ | ------------------------- | ------------------------------- |
> | **OpenCLIP ViT-B/32**   | Vision | 12 | 768 | 224×224 px (49 spatial tokens + 1 CLS) | LAION dataset | 1.4B image–text pairs | Contrastive supervision |
> | **Pythia-deduped-160M** | Language | 12 | 768 | 2048 tokens | The Pile | 300B tokens | Next-token prediction |
> | **Gemma-2B-IT** | Language | 18 | 2048 | 8192 tokens | Google-curated web, code, and math sources | 6T tokens | Next-token prediction |
>
>
> At the output of each transformer layer, the token embeddings $\mathbf{x} \in \mathbb{R}^{d_\mathrm{model}}$​ are known as “residual stream activations” in the LLM explainability literature. These dense embeddings are hard to interpret directly due to entangled (“overlapping”) internal representations. We therefore use sparse autoencoders (SAEs) to map residual activations into a higher-dimensional sparse concept space, $\mathbb{R}^{d_\mathrm{model}} \to \mathbb{R}^{d_\mathrm{SAE}}$ with $d_\mathrm{SAE} \gg d_\mathrm{model}$, i.e., an overcomplete latent space rather than a bottleneck. The encoder applies a linear map followed by ReLU: $f(\mathbf{x}) = \mathrm{ReLU}(F\mathbf{x})$. The entries of $f(\mathbf{x})\geq 0$ constitute the concept activation vector: component $f_i$ reports the presence and magnitude of concept $i$ contained in the activations $\mathbf{x}$. The linear decoder $D$ reconstructs the activation, $\hat{\mathbf{x}} = Df(\mathbf{x}) + \mathbf{b}$, whose columns $\mathbf{d}_i$ define the concept directions. We train SAEs end-to-end with the loss
>
> $$
> \mathcal{L} = ||\mathbf{x} - \hat{\mathbf{x}}||^2_2 + \lambda ||f(\mathbf{x})||_1,
> $$
>
> where the first term ensures accurate reconstruction of the original input activation and the $L_1$ term (tunable hyperparameter $\lambda$) encourages sparsity in the concept space. This yields a representation in which each residual activation is expressed as a sparse linear combination of learned concepts. (Shapes: encoder $F \in \mathbb{R}^{d_\mathrm{SAE} \times d_\mathrm{model}}$, decoder $D \in \mathbb{R}^{d_\mathrm{model} \times d_\mathrm{SAE}}$, bias term $\mathbf{b}\in \mathbb{R}^{d_\mathrm{model}}$)
>
> To address our goal of unpacking the internals of our transformer models (cf. table above), each SAE is trained independently on a single transformer layer’s residual stream activations—e.g., for a 12 layer transformer, we train 12 **distinct** SAEs on the residual stream activations output from the corresponding layer. In this work, we train 3 sets of SAEs, and source one set of pre-trained SAEs from Hugging Face.
>
> Below, we provide the training and evaluation details for each set of SAEs. Specific hyperparameters for training all SAEs can already be found in Appendix A.
>
> *Data used to fit SAEs*
>
> [Section 3]: Concepts identified in noise activations from OpenCLIP ViT-B/32. SAEs trained on 1.3M i.i.d. Gaussian noise image (224×224) activations, validation on 50k ImageNet-1k images, 12 SAEs for all 12 layers. $d_\mathrm{SAE}=49,152$.
>
> [Section 4]: More concepts from transformer activations of increasingly disordered inputs. OpenCLIP ViT-B/32—trained on 1.3M ImageNet-1k activations, validation on 50k ImageNet-1k activations with shuffled patches of various sizes, 12 SAEs for all 12 layers. Pythia-deduped-160M—trained on 204.8M FineWeb-Edu tokens of normal text, validation on 5.12M FineWeb-Edu tokens with shuffled $n$-grams of varying $n$, 12 SAEs for all 12 layers. $d_\mathrm{SAE}=24,576$.
>
> [Section 5]: Hallucination prediction and suppression from concepts identified in activations of Gemma-2B-IT. SAEs pre-trained on 1.23B token activations from FineWeb, validation on 1,006 Vectara benchmark document activations, pre-trained SAEs sourced from Hugging Face for layers $1,7,11,13,17$. $d_\mathrm{SAE}=16,384$.
>
> We appreciate the careful review. We believe that this compact description of our main modeling pipeline will improve reader clarity.

---

> > ### Comment · Reviewer_LyaA · 2025-08-07
> > **Thanks**
> >
> > Thanks. I will be updating my review.

---

> > > ### Author Response · Authors · 2025-08-08
> > >
> > > We thank reviewer LyaA for confirming that they will be updating their review and being open to reconsidering their initial score.

---

> > > ### Author Response · Authors · 2025-08-09
> > >
> > > We wish to thank again the reviewer for engaging in discussion and providing feedback. If our answers to your questions were satisfactory we would be extremely grateful if you could consider increasing your score before the deadline. Please feel free to ask additional questions, should time allow, we would be more than happy to provide further clarifications.

---

### Decision · Program_Chairs · 2025-09-17

**Decision:**

Accept (poster)

**Comment:**

The paper explores how hallucinations arise in transformer models by using sparse autoencoders (SAEs) to analyze internal activations under input uncertainty. The core finding is that under input uncertainty, transformers tend to activate "input-agnostic conceptual representations" which are robustly triggered even by pure noise, thereby increasing the likelihood of hallucinated output.

The authors' rebuttal had convinced the reviewers towards positive final ratings (5/4/5/4). This work is considered a valuable contribution to the field of mechanistic interpretability and AI safety, and is recommended for acceptance.